# Curl Descent: Non-Gradient Learning Dynamics with Sign-Diverse Plasticity

**Hugo Ninou**
Département d'Études Cognitives
École normale supérieure - PSL
hugo.ninou@ens.fr

**Jonathan Kadmon**
Edmond and Lily Safra Center for Brain Sciences
The Hebrew University of Jerusalem
jonathan.kadmon@mail.huji.ac.il

**N. Alex Cayco-Gajic**
Département d'Études Cognitives
École normale supérieure - PSL
natasha.cayco.gajic@ens.fr

## Abstract

Gradient-based algorithms are a cornerstone of artificial neural network training, yet it remains unclear whether biological neural networks use similar gradient-based strategies during learning. Experiments often discover a diversity of synaptic plasticity rules, but whether these amount to an approximation to gradient descent is unclear. Here we investigate a previously overlooked possibility: that learning dynamics may include fundamentally non-gradient "curl"-like components while still being able to effectively optimize a loss function. Curl terms naturally emerge in networks with inhibitory-excitatory connectivity or Hebbian/anti-Hebbian plasticity, resulting in learning dynamics that cannot be framed as gradient descent on *any* objective. To investigate the impact of these curl terms, we analyze feedforward networks within an analytically tractable student-teacher framework, systematically introducing non-gradient dynamics through neurons exhibiting rule-flipped plasticity. Small curl terms preserve the stability of the original solution manifold, resulting in learning dynamics similar to gradient descent. Beyond a critical value, strong curl terms destabilize the solution manifold. Depending on the network architecture, this loss of stability can lead to chaotic learning dynamics that destroy performance. In other cases, the curl terms can counterintuitively speed learning compared to gradient descent by allowing the weight dynamics to escape saddles by temporarily ascending the loss. Our results identify specific architectures capable of supporting robust learning via diverse learning rules, providing an important counterpoint to normative theories of gradient-based learning in neural networks.

## 1 Introduction

Modern deep learning relies on backpropagation to compute high-dimensional gradients that assign credit to individual synapses by propagating error signals backward through the network [Rumelhart et al., 1986, Chinta and Tweed, 2012]. This mechanism solves the credit assignment problem by assuming that each synapse can locally adapt in proportion to the negative gradient of the loss. However, despite its central role in artificial systems, direct evidence for gradient-based learning in biological neural circuits is still lacking [Lillicrap et al., 2020].

To reconcile this gap, many studies have proposed biologically plausible approximations to gradient descent [Richards and Kording, 2023]. These typically address concerns related to the locality of

39th Conference on Neural Information Processing Systems (NeurIPS 2025).

error information [Golkar et al., 2023, Keller and Mrsic-Flogel, 2018, Bredenberg et al., 2023], separation of forward and backward passes [Guerguiev et al., 2017, Xie and Seung, 2003], and the weight transport problem [Akrout et al., 2019, Lillicrap et al., 2016]. However, a more fundamental constraint has received less attention: even if local gradient information were available, it remains unclear whether biological plasticity rules can consistently drive synapses along such a gradient.

Backpropagation implicitly assumes a coordinated update rule in which all synapses adjust their weights in directions aligned with the descending gradient of a global error signal. Yet this assumption is incompatible with experimental observations. Synaptic plasticity in the brain is remarkably diverse, with different cell types and circuits expressing distinct, and sometimes opposing, forms of long-term potentiation and depression, including both Hebbian and anti-Hebbian rules [Abbott and Nelson, 2000]. This diversity is compounded by Dale's law [Dale, 1934], which fixes each neuron as either excitatory or inhibitory, constraining the sign of its outgoing synapses. Crucially, the local plasticity rule at a given synapse appears uncorrelated with the identity of the presynaptic neuron—whether excitatory or inhibitory—or with any other structural or physiological feature of the circuit [Citri and Malenka, 2008]. As a result, identical local signals can produce opposite weight changes across different synapses, with no apparent mechanism to coordinate or compensate for this variability. These constraints raise a more fundamental question: can networks with heterogeneous and potentially antagonistic plasticity rules still support meaningful optimization?

In this work, we examine how physiological diversity in synaptic plasticity and cell-type identity, shapes the learning dynamics of neural networks. Specifically, we ask whether networks composed of heterogeneous neurons can still effectively reduce an objective function or whether such diversity precludes gradient-based learning altogether.

**Our contributions are as follows:**

- We show that non-gradient terms naturally arise in biologically plausible networks due to sign-diverse plasticity. In contrast to many previously considered alternatives to backpropagation, here the learning dynamics cannot be written as gradient descent on *any* objective due to the existence of non-gradient "curl"-like terms.
- We develop a theoretical framework to isolate and systematically analyze the effect of curl terms in large linear feedforward networks. Leveraging random matrix theory, we identify a dynamical phase transition in which the zero-error solution manifold loses stability.
- We demonstrate that the location of this phase transition depends on architectural parameters, particularly the expansion ratio between the input and hidden layers.
- Finally, we provide numerical evidence that, in certain *nonlinear* architectures, curl descent can accelerate learning, even in the absence of true gradient flow.

Our results suggest a previously unexplored mechanism through which biological learning rules could give rise to fundamentally non-gradient dynamics that still support effective learning.

## 2 Non-gradient terms arise in biologically plausible neural networks

Our curl descent learning rule introduces non-gradient terms into the learning dynamics by flipping the *sign* of the gradient descent update for selected synapses. To motivate this approach, in this section we demonstrate how non-gradient learning dynamics can arise from sign-diverse plasticity in biological networks. We return to the curl descent rule in the following section.

Unlike artificial networks, neurons in the brain exhibit a variety of plasticity mechanisms and physiological properties that directly influence how a synapse is updated in relation to the local gradient. We begin our analysis by showing that *sign* diversity in effective learning rules — whether from plasticity mechanisms or the neural dynamics of excitatory-inhibitory networks — gives rise to provably non-gradient terms in the learning dynamics. While we focus on recurrent linear architectures for brevity, it is straightforward to extend this analysis to nonlinear dynamics.

**Excitatory-inhibitory (E-I) networks.** Non-gradient terms can emerge in recurrent E-I Hebbian networks. Consider a linear recurrent neural network (RNN) described by:

$$\tau_y \dot{\mathbf{y}} = -\mathbf{y} + WD\mathbf{y} + \mathbf{f}, \quad \text{for } D = \text{diag}(d_1, \ldots, d_N) \text{ with } d_i \in \{+1, -1\}, \tag{1}$$

where $\mathbf{y} \in \mathbb{R}^N$ represents the firing rates of the $N$ neurons, $W \in \mathbb{R}_{\geq 0}^{N \times N}$ denotes non-negative recurrent weight magnitudes, $d_i$ determines whether neuron $i$ is excitatory or inhibitory, $\tau_y > 0$ is a time constant, and $\mathbf{f} \in \mathbb{R}^N$ is an external drive. The Taylor expansion of the neural dynamics at steady state gives $\mathbf{y}^* = (\mathbb{I} + WD + \mathcal{O}(W^2))\mathbf{f}$. Under Hebbian plasticity (with weight decay), we obtain the following learning dynamics:

$$\tau_W \dot{W} = \mathbf{y}\mathbf{y}^\top - \gamma W \approx \mathbf{f}\mathbf{f}^\top + WD\mathbf{f}\mathbf{f}^\top + \mathbf{f}\mathbf{f}^\top DW^\top - \gamma W. \tag{2}$$

Here, $\gamma > 0$ regulates weight decay to prevent unbounded growth [Gerstner and Kistler, 2002]. While the first and last terms can be written as gradients — namely, $\mathbf{f}\mathbf{f}^\top = \nabla_W \operatorname{Tr}(\mathbf{f}^\top W^\top)$ and $\gamma W = \frac{\gamma}{2} \nabla_W \operatorname{Tr}(WW^\top)$ — the two intermediate terms cannot, unless $\mathbf{f}\mathbf{f}^\top D$ is symmetric. However, this symmetry holds *only* when the inhibitory neurons receive no external input; see Supplementary Material A.3 for details. This is the case in previous normative studies that derive excitatory-inhibitory Hebbian networks from a similarity matching objective [Pehlevan et al., 2015] (see Supplementary Material A.4). In the general case, networks that respect Dale's law will therefore include non-gradient terms in their learning dynamics.

**Hebbian/anti-Hebbian networks.** A similar argument can be made for networks where the learning rule of individual synapses are sign-flipped as a result of mixed Hebbian and anti-Hebbian plasticity; see Supplementary Material A.2 for details.

## 3 Curl descent in a student-teacher framework

To better understand the effect of non-gradient terms (here called "curl" terms in analogy with the Helmholtz decomposition), we next turn to an analytically tractable setting in which the learning dynamics of gradient descent is well understood [Saxe et al., 2014, Advani et al., 2020, Goldt et al., 2020, Baldi and Hornik, 1989, Le Cun et al., 1991, Seung et al., 1992]: linear feedforward networks with a single hidden layer.

We adopt a student-teacher framework in which a two-layer teacher network (parametrized by weights $W_1^\star \in \mathbb{R}^{N \times M}$ and $W_2^\star \in \mathbb{R}^N$) maps an input vector $\mathbf{x} \in \mathbb{R}^M$ to scalar output $y \in \mathbb{R}$ via $y = W_2^\star W_1^\star \mathbf{x}$. The student uses the same architecture, and its output is given by $\hat{y} = W_2 \mathbf{h}$, where $\mathbf{h} = W_1 \mathbf{x}$ is the hidden layer activity. The student's goal is to modify its weights $W_1$ and $W_2$ to match the teacher's output $y$ and minimize the quadratic loss $\mathcal{L} = \frac{1}{2} \langle e^2 \rangle$, where $e := \hat{y} - y$ is the signed error and $\langle \cdot \rangle$ denotes an average over the input distribution.

Standard gradient descent gives the following updates:

$$\Delta W_1^{\text{grad}} = -\nabla_{W_1} \mathcal{L} = W_2^\top W_2 W_1 \mathbf{x}\mathbf{x}^\top - W_2^\top y \mathbf{x}^\top = -W_2^\top e \mathbf{x}^\top \tag{3}$$

$$\Delta W_2^{\text{grad}} = -\nabla_{W_2} \mathcal{L} = -W_2 \mathbf{h}\mathbf{h}^\top + y \mathbf{h}^\top = -e \mathbf{h}^\top \tag{4}$$

Each term is the outer product of a postsynaptic error signal and the presynaptic activity, and can be considered a supervised "Hebbian-like" learning rule [Melchior et al., 2024, Refinetti et al., 2021].

**Curl descent rule.** To model the diverse behavior of plasticity rules observed in biological neural networks, we flip the sign of a subset of synapses using diagonal matrices $D_1 \in \mathbb{R}^{M \times M}$ and $D_2 \in \mathbb{R}^{N \times N}$:

$$\Delta W_1^{\text{curl}} = -W_2^\top e \mathbf{x}^\top D_1, \qquad \Delta W_2^{\text{curl}} = -e \mathbf{h}^\top D_2 \tag{5}$$

where $D_1 = \operatorname{diag}(d_{1,1}, \ldots, d_{1,M})$, $D_2 = \operatorname{diag}(d_{2,1}, \ldots, d_{2,N})$, and $d_{l,j} \in \{+1, -1\}$. Therefore, all synapses associated with presynaptic neuron $j$ in layer $l$ follow either an unchanged learning rule ($d_{l,j} = +1$) or a flipped learning rule ($d_{l,j} = -1$). If both types of learning rule are present in the student, no scalar potential function exists whose gradient reproduces the weight updates (see Supplementary Material). Thus, the addition of rule-flipped plasticity induces intrinsically non-gradient curl terms in the learning dynamics.

## 4 Analytical results

Following previous work on the learning dynamics of linear networks [Saxe et al., 2014], we assume whitened inputs $\langle \mathbf{x}\mathbf{x}^\top \rangle = \mathbb{I}_M$ and take the continuous time limit (small learning rate) giving the

following nonlinear dynamical system for the weights:

$$\dot{W}_1 = W_2^\top(s - W_2W_1)D_1 \quad \text{and} \quad \dot{W}_2 = (s - W_2W_1)W_1^\top D_2, \tag{6}$$

where $s := W_2^\star W_1^\star$ represents the effective function implemented by the teacher. If $D_1 = \mathbb{I}_M$ and $D_2 = \mathbb{I}_N$, the learning dynamics reduce to gradient flow.

Since curl descent only changes the *sign* of the plasticity rule, it will have the same fixed-point solutions as gradient descent: these include a continuous manifold corresponding to $W_2W_1 = s$ (here called the *solution manifold*) plus a discrete fixed point at the origin ($W_1 = W_2 = 0$). In gradient descent, the hyperbolic solution manifold is known to be stable, whereas the origin is a saddle [Saxe et al., 2014]. However, curl terms can have a significant impact on the learning dynamics by changing the stability properties of fixed points. Flipping the sign of all the synapses would have a clearly disastrous impact as it would lead the weights to ascend the gradient. Surprisingly, however, the stability of the solution manifold can be robust to moderate amounts of rule-flipped plasticity depending on the network architecture, as we will demonstrate below.

## 4.1 Toy example: A two-neuron network

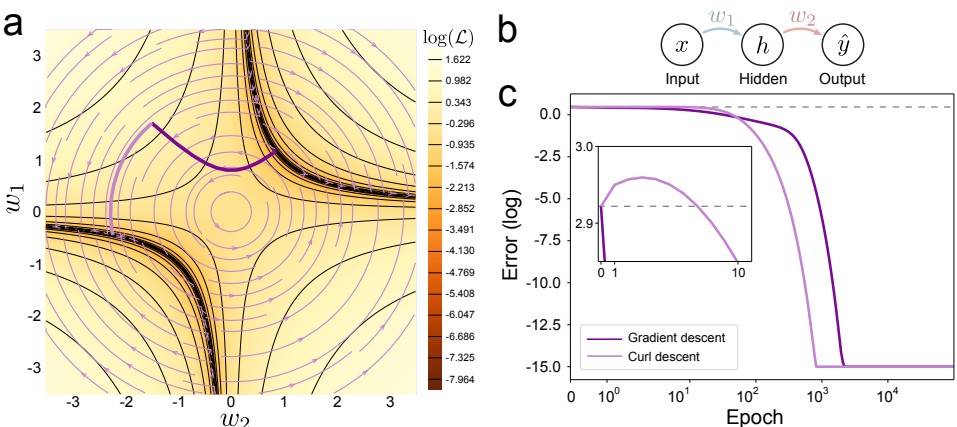

Figure 1: **Toy model analysis. a)** Learning trajectories in weight space for gradient flow (dark purple curve) and curl flow (light purple curve). The heatmap represents the log loss, which determines the gradient descent dynamics. The hyperbolic solution manifold (dark red curves) is a global minimum. Curl descent reshapes the learning dynamics and adds a rotational field (flow-field overlain in light purple curves with arrows). **b)** Schematic of the toy model network. **c)** Log error vs. training epoch for the same learning trajectories shown in panel a. Inset: Same figure zoomed on the first 10 epochs, showing that curl descent initially ascends the loss function.

We first build intuition by considering a minimal two-neuron network ($M = N = 1$). We will denote the resulting scalar weights of the teacher and student as $w_l^\star$ and $w_l$, for $l = 1, 2$. The continuous-time gradient flow dynamics are given by:

$$\dot{w}_1 = w_2(s - w_2w_1), \qquad \dot{w}_2 = w_1(s - w_2w_1). \tag{7}$$

Flipping the sign of the hidden neuron's plasticity rule instead yields the following dynamics:

$$\dot{w}_1 = w_2(s - w_2w_1), \qquad \dot{w}_2 = -w_1(s - w_2w_1) \tag{8}$$

How does this sign flip modify the stability of the solution manifold? To test this we analyze the change of stability of fixed points by calculating the curl flow Jacobian:

$$J = \begin{bmatrix} -w_2^2 & (s - 2w_2w_1) \\ (2w_2w_1 - s) & w_1^2 \end{bmatrix} \tag{9}$$

On the solution manifold ($s = w_1w_2$), the eigenvalues are $\lambda_1 = 0$ and $\lambda_2 = w_1^2 - w_2^2$. Hence, only the fixed points satisfying $|w_2| > |w_1|$ will remain (neutrally) stable, while the other half of the solution manifold loses stability.

The origin, a saddle under the original gradient flow dynamics, is converted to a center with purely imaginary eigenvalues $\lambda_{1,2} = \pm is$ (Fig. 1a). The phase plane shows two qualitatively distinct dynamical regimes: convergence to a minimum on the solution manifold or small-amplitude oscillations, depending on the initialization of the weights. These regimes are separated by heteroclinic orbits on the circle $w_1^2 + w_2^2 = s$.

Under curl flow, the weights evolve according to rotational dynamics induced by the curl terms, but can still descend the loss and converge to the solution manifold. Intriguingly, in some cases curl flow can lead to *faster* convergence compared to gradient flow (Fig. 1). In particular, this happens when the weights would otherwise be stuck along the stable manifold of the saddle at the origin. This hints at a possible benefit of curl terms in helping weight dynamics escape saddles, but comes at a significant cost: half of the solution manifold has lost stability, and small-amplitude initializations no longer converge. This can partially be explained by the fact that in this two-neuron network, we flipped the sign of an "entire layer" to add a curl term. In the following section, we consider large networks where we have finer control over the magnitude of the curl terms.

## 4.2  Large networks

Returning to the general case (arbitrary $M$, $N$), we derive the Jacobian $J$ of the learning dynamics at the origin and on the solution manifold. The eigenvalues of the Jacobian matrix at a given point in parameter space reveal the local stability at this point: if all eigenvalues' real parts are negative, then the point is stable, otherwise it is unstable. We thus analyze the eigenvalues of the Jacobian as we systematically increase the fraction of rule-flipped neurons in either the hidden layer or the readout layer. For this, we quantified by the fraction of negative diagonal elements in $D_1$ or $D_2$ as:

$$\alpha_h = \frac{1}{M}\sum_{j=1}^{M}\mathbf{1}\{d_{1,j} < 0\}, \qquad \alpha_r = \frac{1}{N}\sum_{j=1}^{N}\mathbf{1}\{d_{2,j} < 0\} \tag{10}$$

where $\mathbf{1}$ denotes the indicator function (derivation details can be found in the Supplementary Material).

**Spectrum at the origin.**  At the origin, the characteristic polynomial can be derived as:

$$\det(J - \lambda I) = (-\lambda)^{MN-N}\prod_{i=1}^{N}(\lambda^2 - d_{2,i}\Sigma), \qquad \Sigma := \sum_{j=1}^{M} d_{1,j}s_j^2 \tag{11}$$

Therefore, the origin can have at most $2N$ nonzero eigenvalues: $\lambda = \pm\sqrt{d_{2,i}\Sigma}$. In the case of gradient flow, all $d_{l,j} = 1$ and $\Sigma > 0$, yielding purely real eigenvalues of positive and negative sign (a saddle). If we now add rule-flipped plasticity in the readout layer, then every hidden neuron whose plasticity is sign-flipped (i.e., for each $d_{2,i} = -1$) will convert two of those eigenvalues to be purely imaginary, turning one of the $N$ planes with embedded saddle dynamics to a center point. Instead, if we add sufficient rule-flipped plasticity to the hidden layer (enough so that $\Sigma < 0$), all $N$ of the saddle planes will turn into centers. This strong dependency on the layer foreshadows the important role that the network architecture will play in determining how curl flow impacts learning dynamics.

**Spectrum on the solution manifold.**  Using the Schur complement, we can derive the characteristic polynomial of $J$ evaluated on the solution manifold ($W_1 W_2 = s$) as:

$$\det(J - \lambda I) = (-1)^{NM+N}\lambda^{MN+N-M}\det(\lambda\mathbb{I}_M + \|W_2\|^2 D_1 + W_1^\top D_2 W_1). \tag{12}$$

Hence, at most $M$ eigenvalues are nonzero and their values will be governed by the determinant on the right-hand side in (12). In the case of gradient flow, this reduces to the characteristic polynomial of a second matrix:

$$\det(\lambda\mathbb{I}_M + A) = 0, \quad A = \|W_2\|^2\mathbb{I}_M + W_1^\top W_1 \tag{13}$$

Since $A$ is positive definite, the eigenvalues of $-A$, and hence the nonzero eigenvalues of $J$, are negative, demonstrating that the solution manifold is stable under gradient flow.

How does the stability of the solution manifold change under curl flow? If we flip the sign of all neurons ($\alpha_h = \alpha_r = 1$), all eigenvalues will flip their sign in correspondence. However, there may be intermediate values of $\alpha_h$ or $\alpha_r$ before the solution manifold loses stability. Indeed, we can directly infer from the structure of Eq. (12) that stability depends on two factors: the ratio between

the variances of $W_1$ and $W_2$, given by the ratio $M/N$, and the fraction of flipped synapses, either $\alpha_h$ or $\alpha_r$, depending on the layer being modified. This simplification arises because the stability is determined by the point at which the largest eigenvalue is zero and does not rely on other properties of the eigenvalue distribution.

The characteristic polynomial is difficult to evaluate in general, but we can leverage random matrix theory to predict when the bounded support of the eigenvalue distribution crosses zero in large networks. Here we use the i.i.d. random distribution of teacher weights, and assume that the student weights share the same statistics on the solution manifold [He et al., 2015], namely:

$$(W_1)_{ij} \overset{\text{i.i.d.}}{\sim} \mathcal{N}(0, 1/M) \quad \text{and} \quad (W_2)_{ij} \overset{\text{i.i.d.}}{\sim} \mathcal{N}(0, 1/N). \tag{14}$$

We consider the infinitely wide limit ($M, N \to \infty$) with a fixed "compression" ratio $c := M/N$, allowing us to characterize how the stability properties change as a function of the network architecture. First, we ask how the spectrum of the Jacobian changes as we vary $\alpha_h$ (keeping $\alpha_r = 0$). In this case, we can derive a fourth-order polynomial whose double roots provide the endpoints of the spectral support (see Supplementary Material). The double roots can be solved numerically to obtain the stability boundary as a function of $c$ (Fig. 2, top). The stability boundaries for the complementary case, in which we instead vary $\alpha_r$ (keeping $\alpha_h = 0$), can be found using the method of [Kumar and Sai Charan, 2020] (Fig. 2, bottom).

**Curl-induced destabilization depends on network architecture.**
The phase diagrams in Figure 2 highlight a clear trend: expansive networks ($c < 1$) are inherently more robust to curl terms. When added to the hidden layer, curl flow destabilizes the solution manifold once the compression ratio exceeds $c \approx 0.3$ (Fig. 2, top). This critical value depends only weakly on $\alpha_h$. In contrast, for the readout layer (Fig. 2, bottom), the stability of the solution manifold depends strongly on the magnitude of the curl terms. In particular, at most half of the readout layer can obey rule-flipped plasticity before stability is lost. These results demonstrate the conditions under which curl descent may converge to the same solution manifold as gradient descent. To understand how the learning dynamics change at the stability boundary, we turn to simulations.

# 5 Simulations

To test our theoretical results on the stability of the solution manifold, we simulated networks with a total of $M + N = 220$ neurons, while varying the compression ratio $c$ and the fraction of rule-flipped neurons (either $\alpha_r$ or $\alpha_h$). The hidden and read-out weights of the teacher were sampled i.i.d. from zero-mean distributions, with variance scaled by the number of input neurons to each layer, ensuring that stability depends only on the compression ratio $c = M/N$ and not on the statistics of the weights. The student networks had identical architectures to the teacher networks, with weights initialized from the same distribution (unless otherwise specified). Inputs were sampled as $x_i \overset{\text{i.i.d.}}{\sim} \mathcal{N}(0, 1/\sqrt{2})$, and along with the teacher's outputs, provided the training data for the students. Weight updates were made on the whole training set ($N_{\text{train}} = 250$ samples) with a learning rate $0.1/N_{\text{train}}$ over $N_{\text{epochs}} = 10^5$ epochs. To ensure numerical stability, $W_1$ and $W_2$ were re-normalized at every epoch to match their initial Frobenius norm. When analyzing the stability regimes, we focused on linear networks to be able to compare directly to theory; however, we note that the qualitative properties also extend to nonlinear networks (see Supplementary Material C). Furthermore, in our final results on convergence speed in curl descent we implemented nonlinear networks with tanh activation functions.

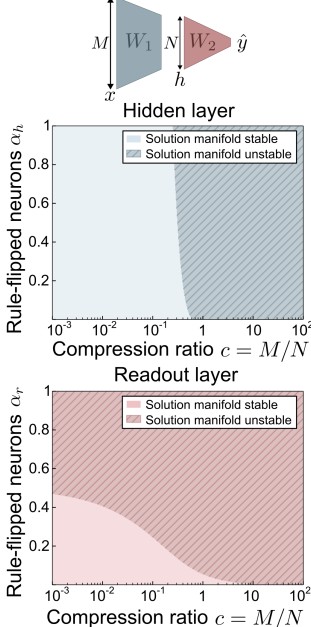

Figure 2: **Analytical phase diagrams.** Stability of the solution manifold as a function of the compression ratio $c$ and the fraction of rule-flipped neurons in each layer $\alpha_h$ (hidden) and $\alpha_r$ (readout).

**Dynamics beyond stability.** In our analytical results, we have shown that above the critical values of $c$ and $\alpha_h$ or $\alpha_r$, the solution manifold loses its stability. What happens to the weight dynamics in this case? Interestingly, this depends on which layer we are flipping: hidden or readout.

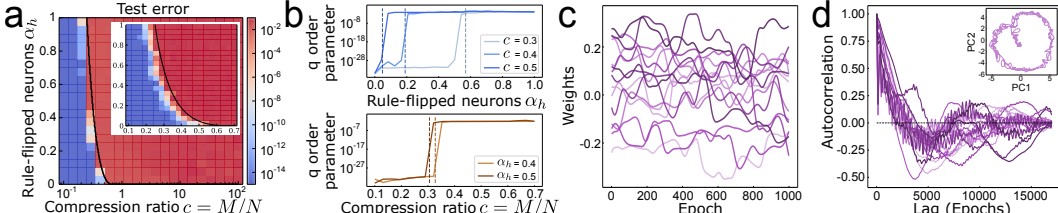

Figure 3: **Hidden layer curl terms lead to chaos. a)** Test error as a function of the compression ratio $c$ and the fraction of rule-flipped neurons $\alpha_h$ (averaged over 10 random seeds). Black curve: analytical stability boundary (cf. Fig. 2, top). Inset: Close-up for $c \in [0.1, 0.7]$. **b)** Order parameter $q$ (averaged over 10 seeds) plotted for varying $\alpha_h$ (top) and $c$ (bottom). Dashed lines indicate analytical transition to instability. **c)** Example weight dynamics in the unstable regime ($c = 0.8$, $\alpha_h = 0.6$). **d)** Example weight autocorrelation functions. Inset: Weight dynamics projected onto its first two principal components. Compute resources: 4 hours on 500 CPUs (local cluster).

**Rule-flipped plasticity in the *hidden* layer.** Our simulations show that introducing rule-flipped weights in the hidden layer induces chaotic learning dynamics when the solution manifold loses stability (Fig. 3). The emergence of chaotic weight dynamics is analogous to the transition to chaos in large disordered networks [Kadmon and Sompolinsky, 2015]. This can occur when the Jacobian always has at least one unstable direction. Notably, our analysis found this to be the case on the solution manifold. Still, our simulations suggest that the dynamics are everywhere unstable, as can be seen from the order parameter quantifying the mean fluctuations:

$$q = \frac{\mathbb{E}\left[\langle (w - \langle w \rangle)^2 \rangle\right]}{\mathbb{V}\left[\langle w \rangle\right]}, \qquad \text{with } w := (W_l)_{ij} \text{ for the sake of notation} \tag{15}$$

where $\langle \cdot \rangle$ represents an average over epochs, and $\mathbb{E}[\cdot], \mathbb{V}[\cdot]$ represent mean, variance over $(l, i, j)$. Here, the transition to chaos appears even in linear networks, because the learning dynamics themselves are inherently nonlinear [Saxe et al., 2014]. The resulting chaos can be understood as a result of a nonlinear weight update combined with structural (quenched) disorder [Sompolinsky et al., 1988].

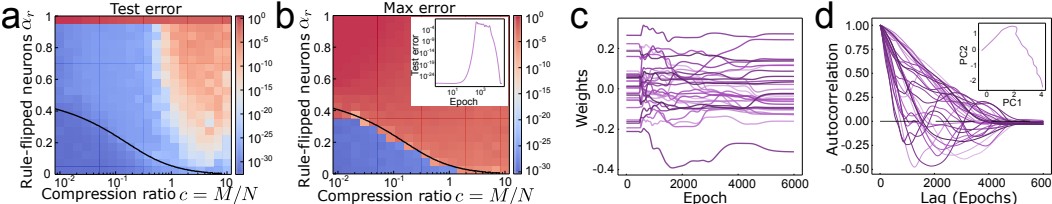

Figure 4: **Readout layer curl terms result in low error even when the solution manifold is unstable. a)** Low test error with readout curl terms. Same as Fig. 3a while varying $\alpha_r$. The black curve shows the analytical stability boundary. **b)** Peak error over learning (maximum over 20 random seeds, initialized near the solution manifold). Inset: Test error vs. epoch in the unstable regime, showing large weight transients that re-descend the loss ($c = 1$, $\alpha_r = 0.6$). **c)** Example weight dynamics in the unstable regime ($c = 1$, $\alpha_r = 0.6$). **d)** Example weight autocorrelation functions. Inset: Weight dynamics projected onto its first two principal components. Compute resources: 4 hours on 500 CPUs (local cluster).

**Rule-flipped plasticity in the *readout* layer.** Surprisingly, destabilizing the solution manifold by introducing rule-flipped plasticity in the readout layer did not necessarily prevent the network from reaching small test error (Fig. 4a). To verify that the solution manifold was indeed unstable, we tried initializing the student networks' weights away from the solution manifold by adding a $10^{-15}$ perturbation. The typical error evolution showed a spike in the loss before going down to another stable minimum (Fig. 4b-d), reminiscent of the dynamics of the two-neuron model in Fig. 1. The dynamics suggest that the solution manifold was indeed unstable, but the weights were nevertheless able to find other low-error regions of the parameter space. We suspect this difference, compared with the chaotic learning dynamics observed when including rule-flipped plasticity in the hidden layer, may be due to the low-dimensional (scalar) output.

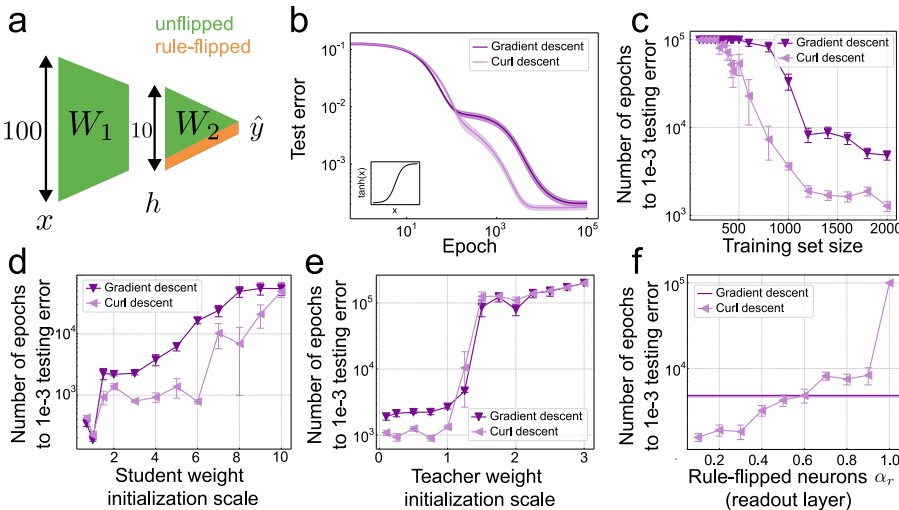

Figure 5: **Nonlinear networks: curl descent leads to faster convergence in a broad parameter regime. a)** Network schematic. **b)** Test error for gradient descent and curl descent with a single rule-flipped readout neuron ($N_{\text{train}} = 2000$, weight initialization scale = 2; error bars indicate mean ± sem, averaged over 10 random seeds). Inset: activation function (tanh). **c)** Convergence speed of curl descent and gradient descent as a function of training set size (weight initialization scale = 2). **d)** Same as c as a function of the weight initialization range ($N_{\text{train}} = 10000$). **e)** Convergence speed as a function of the teacher weights initialization scale. **f)** Convergence speed as a function of the fraction of rule-flipped readout neurons. Compute resources: 12 hours on 500 CPUs (local cluster).

**Faster convergence in nonlinear networks.** These results led us to ask whether curl descent could have any numerical advantage compared to gradient descent. In the toy example, we saw that in some circumstances, curl descent could descend the loss faster than gradient descent (Fig. 1). To test this, we simulated contracting tanh networks with $M = 100$ input units and $N = 10$ hidden units while flipping the learning-rule sign for *a single weight* in the readout layer: i.e., only one of the model's 1010 trainable parameters (Fig. 11a). Indeed, a single flipped weight was able to significantly reshape the learning trajectory, leading to faster convergence (Fig. 11b). This improved performance increased in the high data regime (Fig. 11c, $p < .01$ for $N_{\text{train}} \geq 500$, paired t-test) and was robust to a broad range of student weight initialization scales (Fig. 11d), where $W^{\text{student}}_{\text{rescaled}} = \text{scale} \cdot W^{\text{student}}$.

We next tested whether convergence speed differences generalized to more complex tasks. Using prior results [Poole et al., 2016, Bahri et al., 2020], we modulated the teacher's function complexity by parametrically expanding the initialization range of the teacher weights. We observed a critical value of the teacher weight initialization scale (Fig.11e): above it both curl descent and gradient descent exhibited a sharp reduction in performance with both rules performing similarly poorly, but below it, curl descent consistently outperformed gradient descent ($p < .01$, paired t-test).

Lastly, we investigated the robustness of curl descent's rapid convergence by increasing the proportion of rule-flipped neurons in the readout layer (Fig. 11f). As expected, the speed advantage of curl descent diminished as the fraction of rule-flipped neurons rose, as the resulting learning dynamics became increasingly unstable. However, curl descent was nevertheless able to descend the error faster than gradient descent for $\leq 50\%$ rule-flipped readout weights ($p < .01$, paired t-test).

Further tests will be necessary to fully resolve the impact of curl descent in complex, real-world tasks. Nevertheless, our numerical results demonstrate that in a broad range of hyperparameter settings, curl descent can counterintuitively speed learning by allowing the weight dynamics to find other low-error solutions than those found by gradient descent.

## 6   Related work

**Non-equilibrium neural dynamics.** The curl descent rule induces non-equilibrium learning dynamics by breaking the symmetry of the Jacobian. Such non-gradient systems have also been

studied in the context of complex networks in the absence of plasticity, including in brain dynamics [Nartallo-Kaluarachchi et al., 2025, Daie et al., 2025]. Here, non-reciprocal connections between neurons introduce curl terms that can give rise to a host of different dynamical regimes [Yan et al., 2013, Fruchart et al., 2021]. To our knowledge, this is the first study to systematically investigate non-equilibrium *learning* dynamics.

**Natural gradients.** Natural gradient descent [Amari, 1998] replaces the standard gradient descent update with $\Delta\theta = -\eta\, G^{-1}\nabla_\theta\mathcal{L}$, where the preconditioning matrix $G$ is positive definite. Selecting $G$ equips the parameter space with a Riemannian geometry, determining the learning flow field [Surace et al., 2020]. When $G$ is positive-definite, each step is guaranteed to decrease the objective, yielding monotonic improvement [Richards and Kording, 2023, Richards et al., 2019]. Non-Euclidean metrics have been shown to better reproduce observed weight distributions in the brain [Pogodin et al., 2023]. Curl descent does not induce a Riemannian metric: the corresponding preconditioning matrix is indefinite, possessing both positive and negative eigenvalues. This violates the assumption of natural gradients, producing weight dynamics with qualitatively new behavior, including rotational vector fields and periodic orbits in the parameter space.

These new dynamics are interesting to compare to recent work which argues that a wide class of learning rules can be viewed as natural gradients for a particular metric [Shoji et al., 2024]. The authors give a constructive proof that any rule which monotonically decreases a cost function over some time window implements a form of natural gradients. However, this condition is not satisfied by curl descent: for example, in the toy model in Fig. 1a, small amplitude initial weights result in periodic dynamics. In simulations in larger networks, we did not observe non-decreasing loss curves (as long as only a minority of neurons in the readout layer were flipped), but we cannot a priori rule out the existence of pathological weight initialization regimes that would lead to such behavior.

**Feedback alignment.** A growing body of work uses random projections of the error as a biologically plausible mechanism to circumvent the weight transport problem [Lillicrap et al., 2016, Nøkland, 2016, Refinetti et al., 2021, Hanut and Kadmon, 2025, Clark et al., 2021, Moskovitz et al., 2019, Lindsey and Litwin-Kumar, 2020, Boopathy and Fiete, 2022]. The weight updates are then given by: $\Delta W_1 = -Bex^\top$ and $\Delta W_2 = -eh^\top$, where $B$ is a feedback matrix. If $B = W_2^\top$, we recover gradient descent. Unless $BW_2$ is symmetric, this learning rule cannot be expressed as deriving from a gradient. In classic feedback alignment, $B$ is chosen to be random, therefore, this condition is unlikely to be satisfied (note that $W_2$ has been numerically observed to partially align to $B^\top$ through the weight dynamics [Lillicrap et al., 2020]). The fact that this algorithm offers little control over the non-gradient terms makes it difficult to test their effect systematically. Curl descent enables this control by flipping the learning rule for a randomly selected fraction of neurons.

**Exact learning dynamics.** Our work extends previous analytical studies of exact learning dynamics of gradient descent in linear neural networks [Saxe et al., 2014, Advani et al., 2020, Pellegrino et al., 2023, Bordelon and Pehlevan, 2025, Hanut and Kadmon, 2025]. A contribution of our framework is to devise a tractable learning rule that enables mathematical analysis of how different amounts of non-gradient terms influence weight dynamics. Unlike previous work, our model highlights cases where the dynamics cannot be captured by a potential.

**Normative theories of Hebbian learning.** Decades of work has shown that Hebbian-like learning rules can optimize specified objectives [Bahroun et al., 2023, Melchior et al., 2024, Pehlevan et al., 2015, Pehlevan and Chklovskii, 2019, Tolmachev and Manton, 2020, Hyvärinen and Oja, 1998, Seung and Zung, 2017, Foldiak, 1990, Seung, 2018, Xie and Seung, 2003, Lim, 2021, Obeid et al., 2019, Halvagal and Zenke, 2023, Flesch et al., 2023, Brito and Gerstner, 2016, Lipshutz et al., 2023, O'Reilly, 2001, Eckmann et al., 2024]. [Pehlevan et al., 2015] propose a biologically plausible neural network with Hebbian updates for the excitatory feedforward connections and rule-flipped updates for the lateral inhibitory neurons. The design of this specific network architecture effectively annihilates the curl terms, enabling their Hebbian/anti-Hebbian learning rules to optimize a similarity matching function. Our work demonstrates that in more general architectures, the sign diversity of Hebbian/anti-Hebbain learning rules induces curl terms into the dynamics.

**Excitatory-Inhibitory networks.** Recent work derived learning rules for Dale's law-compliant networks from optimization principles [Cornford et al., 2024]. Other studies have found that excitatory-

inhibitory plasticity enhances memory formation and retrieval in neural networks [Gong and Brunel, 2024, Vogels et al., 2011, Miehl and Gjorgjieva, 2022, Wu et al., 2022, Agnes and Vogels, 2024], without explicitly deriving these rules from a cost function. Our work argues that curl terms originating from the sign diversity inherent in excitatory-inhibitory networks could contribute to improved task performance, using a mechanism distinct from the standard optimization view. In addition, it proposes constraints on the architectures that can support gradient learning with inhibitory neurons.

# 7  Discussion

Here, we demonstrated that non-gradient terms arise due to sign diversity in biologically motivated plasticity rules. We further developed a controlled framework to quantify the impact of increasing non-gradient "curl" components in neural learning. Our results show that depending on network architecture, curl terms can generate chaotic learning dynamics, or they can counterintuitively descend a loss function even if the gradient descent solution manifold is no longer stable – in some cases, even converging faster than gradient descent. More broadly, we have argued for a need to investigate how non-gradient learning dynamics may play a role in task performance in neural networks.

We have shown how easily curl terms could arise through sign-diverse cell types (i.e., E-I) or through sign-diverse plasticity rules. However, gradient descent and sign diversity are not inherently incompatible. For instance, cell-specific plasticity rules might simply reflect how gradient-based optimization manifests differently across neurons, compensating for variations in intrinsic properties or connectivity motifs. For example, cell-specific plasticity in E-I networks can implement gradient descent of a similarity matching objective [Pehlevan et al., 2015]. In Supplementary Material A.4, we show that such networks are able to implement gradient flow by choosing a specific architecture that nullifies curl terms. Sign diversity has also been reported within single neurons, depending on the distance of the synapse to the soma [Froemke, 2010, Sjöström and Häusser, 2006]: this effect has been argued to be compatible with specific implementations of gradient descent [Richards et al., 2019]. Similarly, neuromodulator-induced sign-flips can arise as a natural consequence of negative reinforcement in reward-modulated plasticity rules [Frémaux and Gerstner, 2015].

Another possibility that different plasticity rules could represent competition between distinct objectives locally optimized by different neural populations, such as error minimization versus homeostatic regulation. Indeed, previous work has argued that an interplay of diverse plasticity rules may underlie stable network performance [Confavreux et al., 2025, Zenke et al., 2015]. In the curl descent we examine, however, the plasticity rules do not differ in functional form but instead exhibit opposite signs. Each rule-flipped neuron effectively attempts to ascend the gradient, meaning its cost function becomes the negative of the original loss. These neurons can thus be viewed not merely as competitive but as adversarial. From this perspective, it is particularly striking that the learning dynamics remain robust to such adversarial neurons — and, in some cases, are even accelerated by their presence.

Our results rest on several assumptions that could be relaxed in future work. First, we considered only i.i.d. inputs; structured stimuli may alter stability and convergence properties. Second, our analysis focused on the local stability of critical points, whereas a full investigation of the accelerated convergence observed with curl descent will require a detailed treatment of the global nonlinear dynamics. Third, we limited curl rules to presynaptic identity-based sign flips; exploring fine-grained, synapse-specific sign flips, perhaps correlated with the structural or neuromodulatory factors mentioned above, could reveal additional regimes. Fourth, we restricted our study to two-layer feedforward networks with scalar outputs, whereas the recurrent and deeper architectures common in the brain may exhibit qualitatively different curl-induced phenomena.

Despite these limitations, our framework integrates readily with existing learning rules — gradient descent, feedback alignment, and natural-gradient methods — by treating curl terms as a tunable perturbation. Overall, our results challenge the dominant view that effective learning must follow a gradient [Richards and Kording, 2023]. Instead, biologically plausible diversity in plasticity rules may support optimization through intrinsically non-gradient mechanisms. This suggests that what may appear as biological irregularity could in fact be a feature: an evolutionary strategy that leverages non-gradient dynamics for efficient and robust learning. Embracing this perspective could inform new optimization principles and architectures in machine learning, expanding the landscape beyond traditional gradient descent.

---

Code accompanying our paper is available at `https://github.com/caycogajiclab/Curl_Descent`.

## Acknowledgements

This work was funded by a CDSN doctoral scholarship from the ENS (HN), the Agence National de Recherche (ANR-17-726 EURE-0017, ANR-23-IACL-0008), INSERM, and the Gatsby Charitable Foundation (JK).

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

# A    Learning rules that can not be expressed as gradient descent

In this section we provide proofs that certain learning rules cannot be written as the gradient of any objective.

## A.1    Curl descent learning rule

We will first demonstrate that the curl descent learning rule for feedforward networks (Equation 5 in the main text) cannot be written as a gradient. Here we provide a proof by contradiction for the readout layer ($\Delta W_2 = -eh^\top D_2$). An analogous derivation for the hidden layer ($\Delta W_1 = -W_2^\top ex^\top D_1$) shows that it too cannot be obtained as the gradient of any function.

*Proof.* Suppose there exists an energy function $\mathcal{L}_{\text{eff}}(W_2)$ such that $-\nabla_{W_2} \mathcal{L}_{\text{eff}} = -eh^\top D_2$. In that case, the $ij$th element of the gradient is given by:

$$\left[ \frac{\partial \mathcal{L}_{\text{eff}}}{\partial W_2} \right]_{ij} = \left[ eh^\top D_2 \right]_{ij} \tag{16}$$

$$= \left[ (\hat{y} - y) h^\top D_2 \right]_{ij} \tag{17}$$

$$= \left[ W_2 hh^\top D_2 - yh^\top D_2 \right]_{ij} \tag{18}$$

$$= \sum_{k=1}^{N} W_{2,ik} \left[ hh^\top D_2 \right]_{kj} - \left[ yh^\top D_2 \right]_{ij}. \tag{19}$$

Since $\mathcal{L}_{\text{eff}}$ has a continuous second derivative, we may apply Schwarz' theorem:

$$\frac{\partial^2 \mathcal{L}_{\text{eff}}}{\partial W_{2,il} \partial W_{2,ij}} = \frac{\partial^2 \mathcal{L}_{\text{eff}}}{\partial W_{2,ij} \partial W_{2,il}}. \tag{20}$$

Substituting equation 19 in 20, we obtain

$$\frac{\partial}{\partial W_{2,il}} \left( \sum_{k=1}^{N} W_{2,ik} [hh^\top D_2]_{kj} \right) = \frac{\partial}{\partial W_{2,ij}} \left( \sum_{k=1}^{N} W_{2,ik} [hh^\top D_2]_{kl} \right) \tag{21}$$

$$[hh^\top D_2]_{lj} = [hh^\top D_2]_{jl}. \tag{22}$$

Therefore, $hh^\top D_2$ must be a symmetric matrix. However, since $D_2$ is a diagonal matrix, it will rescale the $i$th column of $hh^\top$ (itself symmetric) by the $D_{2,ii}$. This can only result in a symmetric matrix if $D_2 = \pm \mathbb{I}_N$. Therefore, when $D_2$ is sign diverse, there exists no function $\mathcal{L}_{\text{eff}}$ for which the curl descent rule can be written as gradient descent. $\square$

## A.2    Hebbian/anti-Hebbian networks

Consider a linear recurrent neural network (RNN) in which synaptic plasticity can be either Hebbian or anti-Hebbian. The RNN dynamics are given by $\tau_y \dot{\mathbf{y}} = -\mathbf{y} + W\mathbf{y} + \mathbf{f}$, where $\mathbf{y} \in \mathbb{R}^N$ are the firing rates of the $N$ neurons in the network, $W \in R^{N \times N}$ are the recurrent weights, $\tau_y > 0$ is the time constant, and $\mathbf{f} \in \mathbb{R}^N$ is an external drive. We consider a sign-diverse learning rule where any synapse can be either Hebbian or anti-Hebbian:

$$\tau_W \dot{W} = \mathbf{y}\mathbf{y}^\top \odot M, \quad \text{with } M \in \mathbb{R}^{N \times N} \text{ with elements } M_{ij} \in \{+1, -1\} \tag{23}$$

where $\odot$ denotes the Hadamard product. If synaptic changes occur on a much slower timescale than neural dynamics, we can assume that the firing rates reach a steady state. Since individual weights are typically of order $1/\sqrt{N}$ or smaller [Rubin et al., 2017], the steady state can be written as $\mathbf{y}^* = (\mathbb{I} - W)^{-1}\mathbf{f} = (\mathbb{I} + W + \mathcal{O}(W^2))\mathbf{f}$. The weight dynamics are then given by

$$\tau_W \dot{W} \approx (\mathbf{f}\mathbf{f}^\top + \mathbf{f}\mathbf{f}^\top W^\top + W\mathbf{f}\mathbf{f}^\top) \odot M. \tag{24}$$

Here, the first term can be written as the negative gradient of an objective function: $\mathbf{f}\mathbf{f}^\top \odot M = -\nabla_W \text{Tr} \left( -(\mathbf{f}\mathbf{f}^\top \odot M)W^\top \right)$. However, the last two terms cannot be generally written as the gradient

of any function (unless specific assumptions are made on $M$ and $W$). Thus, the sign diversity of the plasticity rule results in learning dynamics that are governed by both gradient and non-gradient terms.

*Proof.* The first term can be expressed as the negative gradient $\mathbf{ff}^\top \odot M = -\nabla_W \mathcal{L}(W)$, where $\mathcal{L}(W) = -\operatorname{Tr}\left((\mathbf{ff}^\top \odot M)\, W^\top\right)$. It will therefore suffice to demonstrate that the second and third terms cannot be written as a gradient. These terms can be grouped together as:

$$F(W) \;=\; (\Sigma W^\top + W\Sigma) \odot M, \tag{25}$$

where we have defined the covariance-like matrix for the inputs as:

$$\Sigma \;:=\; \mathbf{ff}^\top \quad (\text{or } \Sigma := \langle \mathbf{ff}^\top \rangle \text{ for time-varying inputs}). \tag{26}$$

A necessary condition for a dynamical system to define a gradient flow is that its Jacobian matrix is symmetric at every point $W$ [Perko, 2001]. This symmetry condition implies that:

$$\frac{\partial F_{ij}(W)}{\partial W_{k\ell}} \;=\; \frac{\partial F_{k\ell}(W)}{\partial W_{ij}} \quad \forall\, i,j,k,\ell. \tag{27}$$

In the general case that $W$ is not symmetric, this condition reduces to

$$\left(\delta_{jk}\,\Sigma_{i\ell} + \delta_{ik}\,\Sigma_{\ell j}\right) M_{ij} = \left(\delta_{i\ell}\,\Sigma_{jk} + \delta_{ik}\,\Sigma_{\ell j}\right) M_{k\ell}. \tag{28}$$

In particular, setting $i = j = k$ and $\ell \neq i$ gives

$$2\,\Sigma_{\ell i}\, M_{ii} = \Sigma_{\ell i}\, M_{i\ell}, \tag{29}$$

which would force $\Sigma_{\ell i} = 0$ for $\ell \neq i$. Since $\Sigma$ is rank-one, this would require no more than one neuron receives input (or, in the time-varying case, it requires inputs to be uncorrelated). If neither of these assumptions holds, there is no choice of $M$ (which elements are $\pm 1$) can symmetrize the Jacobian. $\square$

### A.2.1 Special case: Gradient flow for symmetric $W$ and homogenous plasticity

We have shown that in the general case, Hebbian/anti-Hebbian networks cannot be written as gradient descent. However, for specific choices of the architecture, the dynamics can follow a gradient. For example, suppose $W = W^\top$. To ensure this holds for all time, the weight dynamics must also be symmetric, which further requires $M = M^\top$. Then, following the logic above, we can derive the following terms for the Jacobian:

$$\frac{\partial F_{ij}(W)}{\partial W_{k\ell}} = (\delta_{j\ell}\,\Sigma_{ik} + \delta_{jk}\,\Sigma_{i\ell} + \delta_{i\ell}\,\Sigma_{jk} + \delta_{ik}\,\Sigma_{\ell j})\, M_{ij}, \tag{30}$$

$$\frac{\partial F_{k\ell}(W)}{\partial W_{ij}} = (\delta_{j\ell}\,\Sigma_{ik} + \delta_{jk}\,\Sigma_{i\ell} + \delta_{i\ell}\,\Sigma_{jk} + \delta_{ik}\,\Sigma_{\ell j})\, M_{k\ell}. \tag{31}$$

Then, the symmetry condition yields two constraints:

- If $i = k$ and $j \neq \ell$, then $\Sigma_{\ell j}\, M_{ij} = M_{i\ell}\, \Sigma_{\ell j}$, so for each row $i$ and any column pair $(j, \ell)$ with $\Sigma_{\ell j} \neq 0$, one must have $M_{ij} = M_{i\ell}$.

- If $j = \ell$ and $i \neq k$, then $\Sigma_{ik}\, M_{ij} = M_{kj}\, \Sigma_{ik}$, so for each column $j$ and any row pair $(i, k)$ with $\Sigma_{ik} \neq 0$, one must have $M_{ij} = M_{kj}$.

In the generic case $\Sigma_{ij} \neq 0$ for all $i, j$, these conditions force $M$ to be either the all-ones matrix $\mathbf{1}\mathbf{1}^\top$ or its negative. Hence, in the case of symmetric matrices, a gradient flow can be realized only for purely Hebbian or purely anti-Hebbian networks. Indeed, if $M = \mathbf{1}\mathbf{1}^\top$, the learning rule in Equation (24) is symmetric, so that any symmetric initial $W(0)$ remains symmetric, and the learning rule can be written as the following gradient descent:

$$\tau_W \dot{W} \approx \Sigma + \Sigma W + W\Sigma \;=\; -\tfrac{1}{2}\,\nabla_W\!\left(-\operatorname{Tr}\!\left(\Sigma W^\top + W\Sigma W^\top\right)\right). \tag{32}$$

If $M = -\mathbf{1}\mathbf{1}^\top$, the minus-sign simply flips the gradient.

## A.3 Hebbian plasticity in excitatory–inhibitory networks

We will next follow a similar logic to demonstrate that the learning rule (Equation 2 in the main text) cannot be written as gradient descent in general:

$$\tau_W \dot{W} = \mathbf{f}\mathbf{f}^\top + W D \mathbf{f}\mathbf{f}^\top + \mathbf{f}\mathbf{f}^\top D W^\top, \tag{33}$$

where $D \in \mathbb{R}^{N \times N}$ is diagonal with $D_{ii} = \begin{cases} +1, & \text{if neuron } i \text{ is excitatory,} \\ -1, & \text{if neuron } i \text{ is inhibitory.} \end{cases}$

*Proof.* As before, define $\Sigma := \mathbf{f}\mathbf{f}^\top$ (or $\langle \mathbf{f}\mathbf{f}^\top \rangle$ for time-varying inputs). The first term in (33) can be written as a negative gradient: $\Sigma = -\nabla_W(-\text{Tr}(\mathbf{f}\mathbf{f}^\top W^\top))$. Since Equation (33) is symmetric, we may restrict attention to $W = W^\top$; otherwise, requiring symmetry of the Jacobian alone would force $\Sigma = 0$, similar as before. To test for a gradient flow, we again inspect the Jacobian symmetry condition, where we now consider the following function corresponding to the second and third terms of the learning dynamics:

$$F(W) = W D \Sigma + \Sigma D W^\top. \tag{34}$$

A direct calculation gives:

$$\frac{\partial F_{ij}}{\partial W_{k\ell}} = \delta_{ik} \, d_\ell \, \Sigma_{\ell j} + \delta_{i\ell} \, d_k \, \Sigma_{kj} + \delta_{jk} \, d_\ell \, \Sigma_{i\ell} + \delta_{j\ell} \, d_k \, \Sigma_{ik}, \tag{35}$$

$$\frac{\partial F_{k\ell}}{\partial W_{ij}} = \delta_{ki} \, d_j \, \Sigma_{j\ell} + \delta_{jk} \, d_i \, \Sigma_{i\ell} + \delta_{\ell i} \, d_j \, \Sigma_{kj} + \delta_{\ell j} \, d_i \, \Sigma_{ki}. \tag{36}$$

In particular, setting $i = k$ and $j \neq \ell$ results in the following constraint:

$$d_\ell \, \Sigma_{\ell j} = d_j \, \Sigma_{\ell j} \quad \forall j, \ell. \tag{37}$$

Thus, whenever $\Sigma_{\ell j} \neq 0$, one must have $d_\ell = d_j$. In the generic case $\Sigma_{ij} \neq 0$ for every pair $(i, j)$, this forces $D = \pm \mathbb{I}_N$ and so eliminates excitatory/inhibitory diversity. Therefore, in general the Hebbian learning rule in excitatory–inhibitory networks cannot correspond to a gradient flow. □

*Remark.* Note however, that if inhibitory neurons receive zero external input (so that corresponding rows and columns of $\Sigma$ vanish), the constraint in Equation (A.3) is satisfied without collapsing $D$ to $\pm \mathbb{I}_N$, and the curl terms are nullified. In Section A.4, we will demonstrate that previous work proposing excitatory-inhibitory circuits that can optimize a similarity matching function [Pehlevan et al., 2015] fall into this category, where the curl terms are eliminated by the choice of structure of the network.

## A.4 Obtaining gradient flow in EI networks by nullifying curl terms

As in [Pehlevan et al., 2015], we consider a neural network made of an excitatory feedforward layer connecting the input to a hidden layer of excitatory neurons, which themselves form a recurrent loop with inhibitory interneurons. This architecture is commonly used for feature extraction in biologically plausible neural networks [Foldiak, 1990, Rubner and Tavan, 1989, Kung et al., 1994, Leen, 1990].

Pehlevan et al. [2015] showed that Hebbian/anti-Hebbian plasticity in such networks can optimize a similarity matching function. Here we take the inverse approach: instead of asking what is the neural architecture and learning rule that can support the optimization of a given cost function, we impose a neural architecture equipped with a Hebbian learning rule and ask what objective it is optimizing.

Grouping neurons by their structural role, the activity of the full network can be written as: $\mathbf{y} = [\mathbf{y}_{\text{FF}}, \mathbf{y}_{\text{RecE}}, \mathbf{y}_{\text{RecI}}]^\top$. Using the framework in Section 2 of the main text, the neural dynamics are given by

$$\tau_y \dot{\mathbf{y}} = -\mathbf{y} + W D \mathbf{y} + \mathbf{f}, \tag{38}$$

where the weight matrix (following the circuit structure in Pehlevan et al. [2015]) can be written in block form as:

$$W = \begin{pmatrix} 0 & 0 & 0 \\ W_{\text{FF} \to \text{RecE}} & 0 & W_{\text{RecI} \to \text{RecE}} \\ 0 & W_{\text{RecE} \to \text{RecI}} & 0 \end{pmatrix} \tag{39}$$

where the 0 matrices correspond to non-existing synapses. The correlation matrix of the input is

$$\Sigma = \begin{pmatrix} \mathbf{f}\mathbf{f}^\top & 0 & 0 \\ 0 & 0 & 0 \\ 0 & 0 & 0 \end{pmatrix} = \begin{pmatrix} \Sigma_{\text{FF}} & 0 & 0 \\ 0 & 0 & 0 \\ 0 & 0 & 0 \end{pmatrix}, \tag{40}$$

as only the excitatory neurons receive external input, and the $D$ matrix is

$$D = \begin{pmatrix} \mathbb{I} & 0 & 0 \\ 0 & \mathbb{I} & 0 \\ 0 & 0 & -\mathbb{I} \end{pmatrix}. \tag{41}$$

Taking the Taylor expansion of the neural dynamics at steady state around $W$ gives $\mathbf{y}^* = (\mathbb{I} - WD)^{-1}\mathbf{f} = (\mathbb{I} + WD + (WD)^2 + (WD)^3 + \mathcal{O}(W^4))\mathbf{f}$. Under Hebbian plasticity, and noticing that $D\Sigma = \Sigma D = \Sigma$ due to the lack of input to inhibitory neurons, we obtain the following weight dynamics:

$$\tau_W \dot{W} = \mathbf{y}\mathbf{y}^\top \approx \Sigma + W\Sigma + \Sigma W^\top + (WD)^2\Sigma + W\Sigma W^\top + \Sigma(DW^\top)^2 \tag{42}$$

$$+ (WD)^3\Sigma + (WD)^2\Sigma W^\top + W\Sigma(DW^\top)^2 + \Sigma(DW^\top)^3. \tag{43}$$

If we take a close look at the structure of each of these terms one-by-one, in comparison to the block structure of $W$ in Equation (A.4), we can observe that many of the terms are effectively nullified as they predict weight updates for synapses that structurally do not exist.

- $\Sigma$ maps onto non-existant FF→FF synapses and is therefore nullified.

- $W\Sigma = \begin{pmatrix} 0 & 0 & 0 \\ W_{\text{FF}\to\text{RecE}}\Sigma_{\text{FF}} & 0 & 0 \\ 0 & 0 & 0 \end{pmatrix}$ maps onto FF→RecE synapses and is maintained.

- $\Sigma W^\top = \begin{pmatrix} 0 & \Sigma_{\text{FF}}^\top W_{\text{FF}\to\text{RecE}}^\top & 0 \\ 0 & 0 & 0 \\ 0 & 0 & 0 \end{pmatrix}$ maps onto nonexistant RecE→FF synapses is nullified.

- $(WD)^2\Sigma = \begin{pmatrix} 0 & 0 & 0 \\ 0 & 0 & 0 \\ W_{\text{RecE}\to\text{RecI}}W_{\text{FF}\to\text{RecE}}\Sigma_{\text{FF}} & 0 & 0 \end{pmatrix}$ maps onto nonexistant FF→RecI synapses and is nullified. Its transpose $\Sigma(DW^\top)^2$, is also nullified.

- $W\Sigma W^\top = \begin{pmatrix} 0 & 0 & 0 \\ 0 & W_{\text{FF}\to\text{RecE}}\Sigma_{\text{FF}}W_{\text{FF}\to\text{RecE}}^\top & 0 \\ 0 & 0 & 0 \end{pmatrix} = \Sigma_{\text{Rec}}$ is the covariance matrix of the recurrent excitatory neuron activities driven by the feedforward input. It too maps onto non-existant RecE→RecE synapses and is nullified.

- $\Sigma(DW^\top)^2 = \begin{pmatrix} 0 & 0 & \Sigma_{FF}W_{\text{FF}\to\text{RecE}}W_{\text{RecE}\to\text{RecI}} \\ 0 & 0 & 0 \\ 0 & 0 & 0 \end{pmatrix}$ maps onto non-existant RecI→FF synapses and is nullified.

- $(WD)^3\Sigma = \begin{pmatrix} 0 & 0 & 0 \\ -W_{\text{RecI}\to\text{RecE}}W_{\text{RecE}\to\text{RecI}}W_{\text{FF}\to\text{RecE}}\Sigma_{\text{FF}} & 0 & 0 \\ 0 & 0 & 0 \end{pmatrix}$ is maintained.

- $(WD)^2\Sigma W^\top = WD\Sigma_{\text{Rec}} = \begin{pmatrix} 0 & 0 & 0 \\ 0 & 0 & 0 \\ 0 & W_{\text{RecE}\to\text{RecI}}W_{\text{FF}\to\text{RecE}}\Sigma_{\text{FF}}W_{\text{FF}\to\text{RecE}}^\top & 0 \end{pmatrix}$ is maintained.

- $W\Sigma(DW^\top)^2 = \Sigma_{\text{Rec}}DW^\top = \begin{pmatrix} 0 & 0 & 0 \\ 0 & 0 & W_{\text{FF}\to\text{RecE}}\Sigma_{\text{FF}}W_{\text{FF}\to\text{RecE}}^\top W_{\text{RecE}\to\text{RecI}}^\top \\ 0 & 0 & 0 \end{pmatrix}$ is maintained.

- $\Sigma(DW^\top)^3 = \begin{pmatrix} 0 & -\Sigma_{\text{FF}}W^\top_{\text{FF}\to\text{RecE}}W^\top_{\text{RecE}\to\text{RecI}}W^\top_{\text{RecI}\to\text{RecE}} & 0 \\ 0 & 0 & 0 \\ 0 & 0 & 0 \end{pmatrix}$ maps onto non-existant RecE→FF synapses and is nullified.

Note that terms of order 0 and 2 are nullified and we are left only with terms of order 1 and 3 in $W$. Collecting the non-nullified terms results in the following weight dynamics:

$$\tau_W \dot{W} \approx W\Sigma + (WD)^3\Sigma + (WD)^2\Sigma W^\top + W\Sigma(DW^\top)^2 \tag{44}$$

Since the update for $W_{\text{RecI}\to\text{RecE}}$ is the transpose of that of $W_{\text{RecE}\to\text{RecI}}$, one can assume that these two matrices will converge to be the transpose of one another, and after collecting terms the update rule can be expressed as:

$$\tau_W \dot{W} \approx -\tfrac{1}{2}\nabla_W \operatorname{Tr}\left((M - \mathbb{I})\Sigma_{\text{rec}}\right). \tag{45}$$

Here, $M = W_{\text{RecI}\to\text{RecE}}W_{\text{RecE}\to\text{RecI}}$ denotes the disynaptic inhibitory feedback from the recurrent excitatory neurons, and $\Sigma_{\text{rec}}$ is the covariance matrix of the recurrent excitatory neuron activities driven by the feedforward input. Minimizing this energy function achieves two principal goals:

1. First, the term $-\operatorname{Tr}(\Sigma_{\text{rec}})$ promotes the maximization of the total variance captured by the excitatory neurons, thereby driving the feedforward weights to perform a PCA-like extraction of high-variance features from the input covariance $\Sigma_{FF}$.

2. Second, the term $\operatorname{Tr}(M\Sigma_{\text{rec}})$ is minimized. The inhibitory feedback matrix $M$ learns the covariance structure of $\Sigma_{\text{rec}}$. This second term will therefore reduce the learning of features already learned.

Overall, this energy function drives the excitatory population to acquire a progressively decorrelated set of features.

This architecture, equipped with a Hebbian learning rule, allows its learning dynamics to be expressed as a gradient flow. This result is consistent with previous work in which this architecture was previously derived from normative principles [Pehlevan et al., 2015, Kung et al., 1994], although the learning rule and objective here are slightly different. However, introducing input-to-inhibitory synapses as well as recurrent excitatory-to-excitatory or inhibitory-to-inhibitory connections would introduce curl terms in the learning dynamics, which could no longer be written as the gradient of any function.

# B  Large networks

In this section, we provide detailed analytical derivations regarding a two-layer linear neural network with $M$ input neurons, $N$ hidden neurons, and scalar output, and whose weights evolve under curl descent (Equation 5 in the main text). In section B.1, we provide an expression of the Jacobian of the weights' dynamics of our system in the general case, and then use this expression to derive its eigenvalues on two types of critical points: the origin saddle (section B.2) and the solution manifold of gradient descent (section B.3). Finally, in the latter case, we leverage random matrix theory (section B.4) to characterize, in the large $M, N$ limit, when the support of the Jacobian's eigenvalues on the solution manifold crosses the origin, characterizing the phase transition from stability (exclusively negative eigenvalues) to instability (including positive eigenvalues).

## B.1  Full derivation of the Jacobian

Consider $W_1 \in \mathbb{R}^{N \times M}$ matrix and $W_2 \in \mathbb{R}^{1 \times N}$ matrix, $D_1$ and $D_2$ two diagonal matrices of respective sizes $M \times M$ and $N \times N$, with diagonal elements $d_{1,i} = \pm 1$ and $d_{2,i} = \pm 1$.

As in the main text, the curl descent learning dynamics are given by:

$$\dot{W}_1 = W_2^\top (s - W_2 W_1) D_1 \tag{46}$$

$$\dot{W}_2 = (s - W_2 W_1) W_1^\top D_2 \tag{47}$$

We will use $E := s - W_2 W_1 \in \mathbb{R}^M$, $\text{vec}(W_1) \in \mathbb{R}^{NM}$ and $\text{vec}(W_2) \in \mathbb{R}^N$. The full Jacobian expression can be broken down to a block matrix with four blocks:

**Full Jacobian expression.**

$$J = \begin{pmatrix} J_{11} & J_{12} \\ J_{21} & J_{22} \end{pmatrix} \tag{48}$$

We proceed to compute the expression of each of these blocks.

$J_{11}$ **computation of** $\frac{\partial \dot{W}_1}{\partial W_1}$.

$$\dot{W}_{1,ij} = W_{2,i} E_j d_{1j} \tag{49}$$

$$\dot{W}_{1,ij} = W_{2,i}(s_j - \sum_k W_{2,k} W_{1,kj}) d_{1j} \tag{50}$$

$$\frac{\partial \dot{W}_{1,ij}}{\partial W_{1,pq}} = -W_{2,i} W_{2,p} \left( d_{1,q} \delta_{jq} \right) \tag{51}$$

$$J_{11} = \frac{\partial \dot{W}_1}{\partial W_1} = -(W_2^T W_2) \otimes D_1 \tag{52}$$

$J_{12}$ **computation of** $\frac{\partial \dot{W}_1}{\partial W_2}$.

$$\frac{\partial \dot{W}_{1,ij}}{\partial W_{2,h}} = -W_{2,i} W_{1,hj} d_{1,j} + \delta_{hi} \left( s_j - \sum_k W_{2,k} W_{1,kj} \right) \tag{53}$$

$$= \delta_{hi} E_j d_{1,j} - W_{2,i} W_{1,hj} d_{1,j} \tag{54}$$

$$[J_{12}]_{(ij),h} = \left[ \frac{\partial \dot{W}_1}{\partial W_2} \right]_{(ij),h} = \delta_{hi} E_j d_{1,j} - W_{2,i} W_{1,hj} d_{1,j} \tag{55}$$

$J_{21}$ **computation of** $\frac{\partial \dot{W}_2}{\partial W_1}$.

$$\dot{W}_{2,\ell} = \sum_{k=1}^{M} E_k W_{1,\ell k} d_{2,\ell} \quad \text{with } E_k = s_k - \sum_t W_{2,t} W_{1,tk} \tag{56}$$

$$\frac{\partial \dot{W}_{2,\ell}}{\partial W_{1,pq}} = \delta_{\ell p} E_q d_{2,\ell} - W_{2,p} W_{1,\ell q} d_{2,\ell} \tag{57}$$

$$[J_{21}]_{\ell,(pq)} = \left[\frac{\partial \dot{W}_2}{\partial W_1}\right]_{\ell,(pq)} = \delta_{\ell p} E_q d_{2,\ell} - W_{2,p} W_{1,\ell q} d_{2,\ell} \tag{58}$$

$J_{22}$ **computation of** $\frac{\partial \dot{W}_2}{\partial W_2}$.

$$\dot{W}_{2,\ell} = \sum_{k=1}^{M} E_k W_{1,\ell k} d_{2,\ell} \quad \text{with } E_k = s_k - \sum_t W_{2,t} W_{2,tk} \tag{59}$$

$$J_{22} = \frac{\partial \dot{W}_{2,\ell}}{\partial W_{2,h}} = -\sum_k W_{1,hk} W_{1,\ell k} d_{2,\ell} = -\left[D_2 W_1 W_1^\top\right]_{\ell h} \tag{60}$$

$$\tag{61}$$

The eigenvalues of this Jacobian are given by $\det(J - \lambda \mathbb{I}) = 0$.

$$J - \lambda \mathbb{I} = \begin{pmatrix} J_{11} - \lambda \mathbb{I}_{NM} & J_{12} \\ J_{21} & J_{22} - \lambda \mathbb{I}_N \end{pmatrix} = \begin{pmatrix} A & B \\ C & D \end{pmatrix} \tag{62}$$

$$\tag{63}$$

## B.2 Evaluating the Jacobian eigenvalues at the origin

At the critical point $(W_1, W_2) = (0,0)$, we have $\dot{W}_1 = 0^\top E D_1$ and $\dot{W}_2 = E 0^\top D_2$, with $E = s - W_2 W_1 = s$. The Jacobian blocks become

$$J_{11} = 0_{NM \times NM} \tag{64}$$

$$[J_{12}]_{(ij),h} = \delta_{hi} s_j d_{1,j} \tag{65}$$

$$[J_{21}]_{l,(pq)} = \delta_{lp} s_q d_{2,l} \tag{66}$$

$$J_{22} = 0_{N \times N} \tag{67}$$

And therefore

$$J(0,0) - \lambda \mathbb{I} = \begin{pmatrix} -\lambda \mathbb{I}_{NM} & J_{12} \\ J_{21} & -\lambda \mathbb{I}_N \end{pmatrix} \tag{68}$$

If $M > 1$, then $NM \neq N$ and $\det(J(0,0) - 0) = 0$, meaning that $\lambda = 0$ is an eigenvalue of the Jacobian.

For the nonzero eigenvalues, the Schur complement yields:

$$\det(J - \lambda \mathbb{I}) = \det(-\lambda \mathbb{I}_{NM}) \det\left(-\lambda \mathbb{I}_N - J_{21}(-\lambda \mathbb{I}_{NM})^{-1} J_{12}\right) \tag{69}$$

$$= (-\lambda)^{NM} \det\left(-\lambda \mathbb{I}_N + \frac{1}{\lambda} J_{21} J_{12}\right) \tag{70}$$

$$= (-\lambda)^{NM-N} \det\left(\lambda^2 \mathbb{I}_N - J_{21} J_{12}\right) \tag{71}$$

$$[J_{21}J_{12}]_{lh} = \sum_{ij} \delta_{li} s_j d_{2,l} \delta_{hi} s_j d_{1,j} \tag{72}$$

$$= d_{2,l} \delta_{lh} \sum_{j=1}^{M} d_{1,j} s_j^2 \tag{73}$$

$$= \left[ \left( \sum_{j=1}^{M} d_{1,j} s_j^2 \right) D_2 \right]_{lh} \tag{74}$$

Hence we have

$$\det(J(0,0) - \lambda \mathbb{I}) = (-\lambda)^{MN-N} \prod_{i=1}^{N} \left( \lambda^2 - d_{2,i} \sum_{j=1}^{M} d_{1,j} s_j^2 \right) \tag{75}$$

Therefore the origin, which is the only saddle point in parameter space, has $MN - N$ zero eigenvalues and $2N$ non-zero eigenvalues:

$$\lambda_i = \pm \begin{cases} \sqrt{\sum_{j=1}^{M} d_{1,j} s_j^2} & \text{if } d_{2,i} = 1 \\ i\sqrt{\sum_{j=1}^{M} d_{1,j} s_j^2} & \text{if } d_{2,i} = -1 \end{cases} \tag{76}$$

The origin is turned into a saddle-center.

## B.3 Evaluating the Jacobian eigenvalues on the solution manifold ($E = 0$)

The Schur complement formula gives $\det(J - \lambda I) = \det(A) \det(D - CA^{-1}B)$ provided that $A$ is invertible. We will investigate the stability of the fixed points of the dynamics verifying $E = 0$.

We compute the determinant of matrix $A = -(W_2^\top W_2) \otimes D_1 - \lambda I_{MN}$ by computing the determinant of each block $A_i = -d_{1,i}(W_2^\top W_2) - \lambda \mathbb{I}_N$ for $i \in [\![1; M]\!]$. Noticing that $W_2^\top W_2$ is rank 1 and applying the matrix determinant lemma results in

$$\det(A_i) = (-1)^N (\lambda + W_2 W_2^\top d_{1,i}) \lambda^{N-1} \tag{77}$$

$$= (-1)^N \lambda^{N-1} (\lambda + d_{1,i} \|W_2\|^2). \tag{78}$$

Therefore

$$\det(A) = \prod_{i=1}^{M} (-1)^N \lambda^{N-1} (\lambda + d_{1,i} \|W_2\|^2) \tag{79}$$

$$= (-1)^{MN} \lambda^{M(N-1)} \prod_{i=1}^{M} (\lambda + d_{1,i} \|W_2\|^2). \tag{80}$$

We now compute the Schur complement $D - CA^{-1}B$, starting with the $A^{-1}$ matrix. Using the Sherman-Morrison formula:

$$A_i^{-1} = \frac{d_{1,i} W_2^\top W_2}{\lambda(\lambda + d_{1,i} \|W_2\|^2)} - \frac{\mathbb{I}_N}{\lambda} \tag{81}$$

Importantly, $W_2^\top$ is an eigenvector of the $A_i^{-1}$ matrices:

$$A_i^{-1} W_2^\top = \frac{-W_2^\top}{\lambda + d_{1,i} \|W_2\|^2}. \tag{82}$$

Therefore simplifying the calculation of $A_j^{-1}B_j$ for one block $j$:

$$\left[A_j^{-1}B_j\right]_{ih} = \frac{W_{2,i}W_{1,hj}}{\lambda + d_{1,j}\|W_2\|^2}d_{1,j} \tag{83}$$

Yielding

$$\left[A^{-1}B\right]_{(ij),h} = \frac{W_{2,i}W_{1,hj}}{\lambda + d_{1,j}\|W_2\|^2}d_{1,j} \tag{84}$$

Multiplying on the left by $[C]_{l,(pq)} = -W_{2,p}W_{1,lq}d_{2,l}$ yields:

$$\left[CA^{-1}B\right]_{l,h} = -\sum_i^N\sum_j^M W_{2,i}W_{1,lj}d_{2,l}\frac{W_{2,i}W_{1,hj}}{\lambda + d_{1,j}\|W_2\|^2}d_{1,j} \tag{85}$$

$$= -\|W_2\|^2 d_{2,l}\sum_j^M W_{1,lj}\frac{d_{1,j}}{\lambda + d_{1,j}\|W_2\|^2}W_{1,jh}^\top \tag{86}$$

$$\tag{87}$$

Finally,

$$\left[D - CA^{-1}B\right]_{lh} = \left[-D_2W_1W_1^\top - \lambda\mathbb{I}_N\right]_{lh} + \|W_2\|^2 d_{2,l}\sum_j^M W_{1,lj}\frac{d_{1,j}}{\lambda + d_{1,j}\|W_2\|^2}W_{1,jh}^\top \tag{88}$$

$$= -\lambda\delta_{lh} + d_{2,l}\sum_{j=1}^M W_{1,lj}\left(\frac{d_{1,j}\|W_2\|^2}{\lambda + d_{1,j}\|W_2\|^2} - \frac{\lambda + d_{1,j}\|W_2\|^2}{\lambda + d_{1,j}\|W_2\|^2}\right)W_{1,jh}^\top \tag{89}$$

$$= -\lambda\delta_{lh} - \lambda d_{2,l}\sum_{j=1}^M W_{1,lj}\frac{1}{\lambda + d_{1,j}\|W_2\|^2}W_{1,jh}^\top \tag{90}$$

$$= -\lambda\left(D_2W_1\Lambda W_1^\top + \mathbb{I}_N\right) \quad \text{with } \Lambda := \mathrm{diag}\left(\frac{1}{\lambda + d_{1,j}\|W_2\|^2}\right) \tag{91}$$

$$\tag{92}$$

And we have, as for the expression of the determinant of $D - CA^{-1}B$:

$$\det(D - CA^{-1}B) = \det\left(-\lambda\left(D_2W_1\Lambda W_1^\top + \mathbb{I}_N\right)\right) \tag{93}$$

$$= (-\lambda)^N\det\left(\mathbb{I}_N + D_2W_1\Lambda W_1^\top\right) \tag{94}$$

$$= (-\lambda)^N\det\left(\mathbb{I}_M + W_1^\top D_2W_1\Lambda\right) \quad \text{as } \det(\mathbb{I} + XY) = \det(\mathbb{I} + YX) \tag{95}$$

$$= (-\lambda)^N\det\left(\left(\Lambda^{-1} + W_1^\top D_2W_1\right)\Lambda\right) \tag{96}$$

$$= (-\lambda)^N\det\left(\Lambda^{-1} + W_1^\top D_2W_1\right)\det\left(\Lambda\right) \tag{97}$$

Noticing that $\det(\Lambda) = (-1)^{MN}\lambda^{M(N-1)}\det(A)^{-1}$, the full determinant of $(J - \lambda\mathbb{I})$ now reads:

$$\boxed{\det\left(J - \lambda\mathbb{I}\right) = (-1)^{NM+N}\lambda^{MN+N-M}\det\left(\mathrm{diag}(\lambda + d_{1,j}\|W_2\|^2) + W_1^\top D_2W_1\right)} \tag{98}$$

The eigenvalues of the Jacobian at the solution points are determined by the equation $\det\left(J - \lambda\mathbb{I}\right) = 0$. One can therefore look at the conditions on the network's architecture parameters $M$ and $N$ such that the solution manifold remains stable upon the introduction of rule-flipped neurons in either the hidden layer or the readout.

## B.4 Evaluating the stability of solution points (Jacobian eigenvalue distribution for $E = 0$)

In the following, we determine the conditions on the architecture parameters $M, N$ and the plasticity parameters conveyed by $D_1, D_2$ needed to ensure the stability of the solution manifold determined by $E = 0$. We will separate two case scenarios: one where $D_1$ has a mix of $\pm 1$ with $D_2 = \mathbb{I}_N$, and inversely where $D_1 = \mathbb{I}_M$ and $D_2$ has a mix of $\pm 1$.

### B.4.1 $D_1$ with mixed signature and $D_2 = \mathbb{I}_M$

In that case, we have from equation 98

$$\det\left(J - \lambda\mathbb{I}\right) = (-1)^{NM+N}\lambda^{MN+N-M}\det\left(\mathrm{diag}(\lambda + d_{1,j}\|W_2\|^2) + W_1^\top W_1\right) \tag{99}$$

The Jacobian therefore has $MN + N - M$ null eigenvalues and the others verify the equation

$$\det\left(\mathrm{diag}(\lambda + d_{1,j}\|W_2\|^2) + W_1^\top W_1\right) = 0 \tag{100}$$

$$\det\left(-\lambda\mathbb{I}_M - \underbrace{\left(\mathrm{diag}(d_{1,j}\|W_2\|^2) + W_1^\top W_1\right)}_{:=X}\right) = 0 \tag{101}$$

That is, we would like to determine the eigenvalue distribution of the above defined $X \in \mathbb{R}^{M \times M}$ matrix which is a sum of a diagonal indefinite matrix with a Wishart matrix.

To proceed, we will assume that:

$$W_{1,ij} \overset{\text{i.i.d.}}{\sim} \mathcal{N}(0, 1/M), \qquad W_{2,i} \overset{\text{i.i.d.}}{\sim} \mathcal{N}(0, 1/N) \tag{102}$$

The $1/M$ and $1/N$ scaling for the $W_1$ and $W_2$ matrices account for He initialization He et al. [2015] and ensures that $W_1 W_1^\top$ has a non-divergent spectrum without any further rescaling.

For large $N$, the law of large numbers gives $\|W_2\|^2 \approx 1$.

Without loss of generality, let

$$D_1 = \mathrm{diag}\big(\underbrace{+1, \ldots, +1}_{m_+}, \underbrace{-1, \ldots, -1}_{m_-}\big), \qquad \alpha_h := \frac{m_-}{M}, \quad \Delta := (1 - \alpha_h) - \alpha_h = 1 - 2\alpha_h \in [-1, 1]. \tag{103}$$

Define the ratio

$$c := \frac{M}{N}. \tag{104}$$

The object of interest is $X := D_1 + W_1 W_1^\top \in \mathbb{R}^{M \times M}$ in the joint limit $M, N \to \infty$ with fixed $c$.

**Cauchy transform of $D_1$.** For $z \notin \{\pm 1\}$,

$$G_{D_1}(z) = \frac{1 - \alpha_h}{z - 1} + \frac{\alpha_h}{z + 1} = \frac{z + \Delta}{z^2 - 1} \tag{105}$$

**Blue transform of $D_1$.** Set $w := G_{D_1}(z)$ and invert:

$$w(z^2 - 1) = z + \Delta \implies wz^2 - z - (w + \Delta) = 0. \tag{106}$$

Solving this quadratic equation for $z$ and choosing the branch with $B_{D_1}(w) \sim 1/w$ as $w \to 0$ yields

$$B_{D_1}(w) = \frac{1 + \sqrt{1 + 4w(w + \Delta)}}{2w} \tag{107}$$

**R-transform of $D_1$.** The R-transform is defined as $R(w) = B(w) - \frac{1}{w}$. Hence

$$R_{D_1}(w) = \frac{\sqrt{1 + 4w(w + \Delta)} - 1}{2w} \tag{108}$$

**Marcenko-Pastur law with variance $1/M$.** Let $S := W_1 W_1^\top$. Note that here the entries have variance $1/M$, and not the usual sample-covariance prefactor $1/N$. Marčenko and Pastur [1967] give the limiting law

$$\mu_S = \mathrm{MP}(c), \qquad \text{support } \left[(1 - \sqrt{c})^2/c, \, (1 + \sqrt{c})^2/c\right] \tag{109}$$

**R-transform of $S := W_1 W_1^\top$.** Using the property of the R-transform $R_{aX}(w) = aR_X(aw)$ with $a = c$ we obtain

$$R_S(w) = \frac{1}{c(1 - w)} \tag{110}$$

**Asymptotic freeness of $D_1$ and $S$.** $S = WW^\top$ is orthogonally invariant, i.e. $USU^\top \overset{d}{=} S$ for any deterministic $U \in O(M)$, because $W$ is Gaussian. An orthogonally invariant random matrix is asymptotically free from any deterministic matrix Collins and Male [2014]. Therefore

$$D \text{ and } S \qquad \text{are } free \tag{111}$$

**Combined R-transform of $X$.** From (108)–(110)

$$R_X(w) = R_{D_1}(w) + R_S(w) = \frac{\sqrt{1 + 4w(w + \Delta)} - 1}{2w} + \frac{1}{c(1 - w)} \tag{112}$$

**Blue transform of $X$.**

$$B_X(w) = \frac{1}{w} + R_X(w) = \frac{1 + \sqrt{1 + 4w(w + \Delta)}}{2w} + \frac{1}{c(1 - w)}. \tag{113}$$

**Implicit equation for the Cauchy transform.** Let $G_X(z)$ be the Cauchy transform of $X$. By definition of the Blue transform:
$$B_X\big(G_X(z)\big) = z \tag{114}$$

Write $\omega := G_X(z)$ and $z = x \in \mathbb{R}$ (real spectral parameter). Introduce

$$A(\omega, x) := 2\omega\left[x - \frac{1}{c(1 - \omega)}\right] - 1, \qquad R(\omega) := 1 + 4\omega(\omega + \Delta) \tag{115}$$

Equation (113) is equivalent to
$$A(\omega, x)^2 = R(\omega). \tag{116}$$
Multiply (116) by $(1 - \omega)^2$ to clear the denominator. Collecting terms produces the quartic polynomial
$$P_x(\omega) := a_4 \omega^4 + a_3 \omega^3 + a_2 \omega^2 + a_1 \omega + a_0 = 0, \tag{117}$$
with coefficients:

$$a_4 = 4(x^2 - 1), \tag{118}$$

$$a_3 = \frac{-4\Delta c - 8cx^2 - 4cx + 8c + 8x}{c}, \tag{119}$$

$$a_2 = \frac{8\Delta c^2 + 4c^2 x^2 + 8c^2 x - 4c^2 - 8cx - 4c + 4}{c^2}, \qquad \text{with } \Delta = 1 - 2\alpha_h \tag{120}$$

$$a_1 = \frac{-4\Delta c - 4cx + 4}{c}, \tag{121}$$

$$a_0 = 0. \tag{122}$$

Differentiating this polynomial gives

$$P'_x(\omega) = 16(x^2 - 1)\omega^3 + \frac{-12\Delta c - 24cx^2 - 12cx + 24c + 24x}{c}\omega^2 \tag{123}$$

$$+ \frac{16\Delta c^2 + 8c^2x^2 + 16c^2x - 8c^2 - 16cx - 8c + 8}{c^2}\omega + \frac{-4\Delta c - 4cx + 4}{c}. \tag{124}$$

with $\Delta = 1 - 2\alpha_h$

**Support of the spectrum**   An endpoint of the support occurs when $\omega$ becomes a double root of (117), i.e.

$$P_x(\omega) = 0, \qquad P'_x(\omega) = 0. \tag{125}$$

Eliminating $\omega$ from (125) yields a quartic polynomial in $x$; its real roots appear pairwise. The eigenvalues support will therefore be the union of at most two intervals.

When the support yields exclusively negative eigenvalues, then the solution manifold is stable. The theoretical boundary for the solution manifold stability corresponds to when the support crosses $0$, that is when the Jacobian on the solution manifold starts having positive eigenvalues.

### B.4.2   $D_2$ with mixed signature and $D_1 = \mathbb{I}_M$

The stability boundaries for the complementary case, in which we instead vary $\alpha_r$ (keeping $\alpha_h = 0$), can be found using the method of [Kumar and Sai Charan, 2020].

# C Simulations for tanh networks

In the main text, we showed that in linear networks, the learning dynamics beyond the stability boundary yielded a transition to chaos when rule-flipped neurons were introduced in the hidden layer, destroying performance. We also showed that when destabilizing the solution manifold by introducing rule-flipped neurons in the readout layer, the network still managed to reach small testing error. In this section, we provide additional simulation results for nonlinear *tanh* networks. The qualitative behavior of the tanh networks is similar to that of the linear ones: we recover the transition to chaos (Figure 7) and the ascend then re-descend mechanism (Figure 9) and the phase diagrams show similar trends although the boundary are shifted to lower $c$ values (figures 6 and 8).

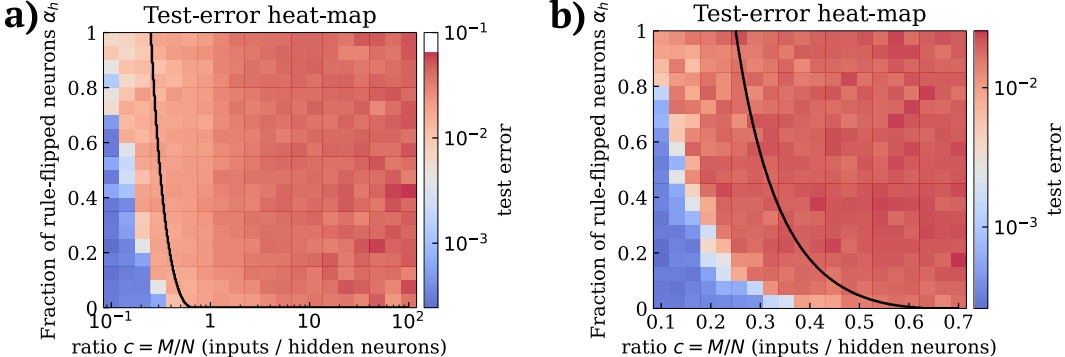

Figure 6: **Hidden layer phase transition for tanh networks. a)** Test error as a function of the compression ratio $c$ and the fraction of rule-flipped neurons $\alpha_h$ (averaged over 10 seeds). Black curve: analytical stability boundary derived for linear networks. **b)** Close-up for $c \in [0.1, 0.7]$. Compute resources: 6 hours on 500 CPUs (local cluster).

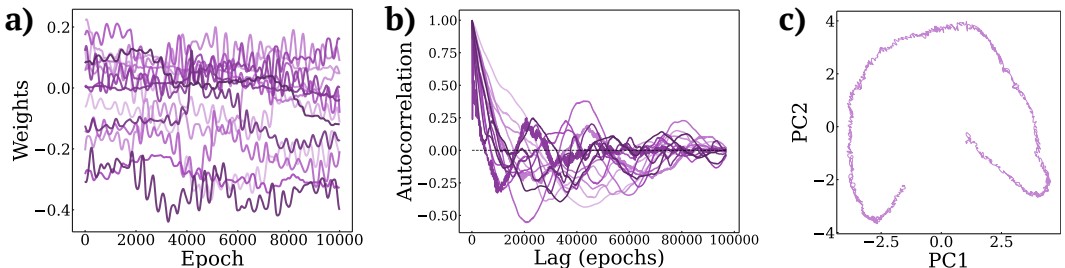

Figure 7: **Hidden layer curl terms lead to chaos in tanh networks.** Example simulation of a tanh network with $N_{\text{tot}} = 110$ neurons, $c = 0.8$ and $\alpha_h = 0.6$. **a)** Weight dynamics as a function of the epochs. **b)** Weight autocorrelation functions. **c)** Weight dynamics projected on its first two principal components.

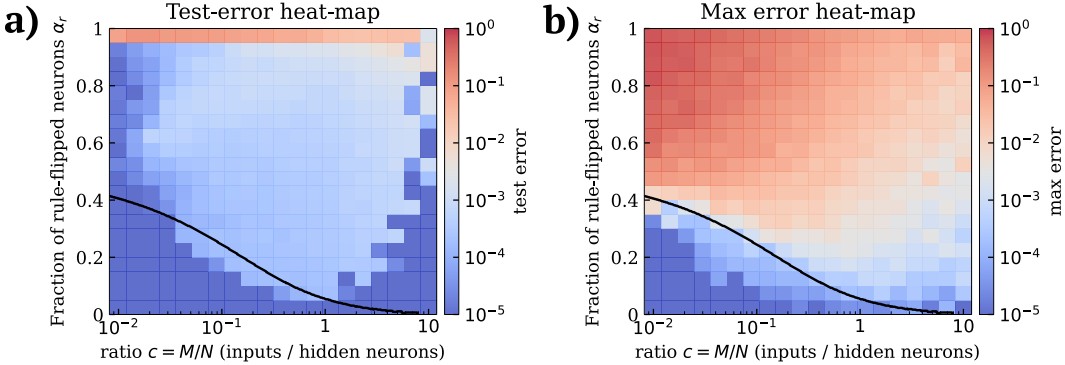

Figure 8: **Readout layer phase transition for tanh networks. a)** Test error as a function of the compression ratio $c$ and the fraction of rule-flipped neurons $\alpha_h$ (averaged over 10 seeds). Black curve: analytical stability boundary derived for linear networks. **b)** Peak learning error (maximum over 20 random seeds, initialized near the solution manifold). The black curve shows the analytical boundary derived in the linear case. Compute resources: 6 hours on 500 CPUs (local cluster).

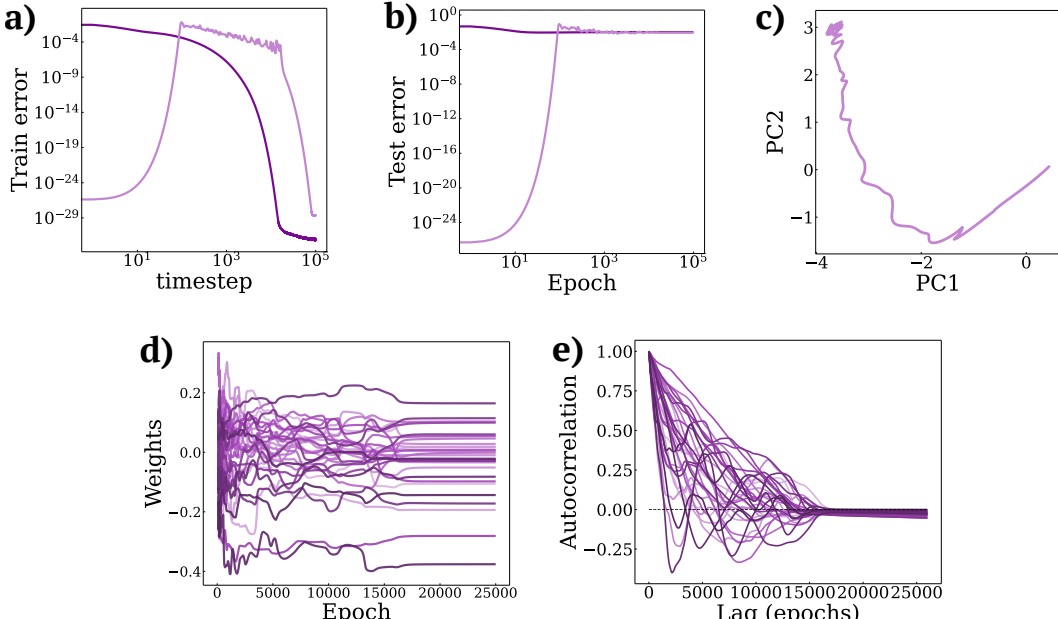

Figure 9: **Readout layer curl terms in tanh networks result in low error even when the solution manifold is unstable.** Example simulation of a tanh network, with $N_{\text{tot}} = 110$ neurons, $c = 1$ and $\alpha_r = 0.6$. **a)** Training error for Curl descent initialized a small distance away from the solution manifold by adding a $10^{-15}$ perturbation on the weights (light purple), and training error for gradient descent, initialized randomly (dark purple). **b)** Same as a for testing error. **c)** Weight dynamics projected on its first two principal components. **d)** Weight dynamics as a function of the epochs. **e)** Weight autocorrelation functions.

# D   Faster convergence for ReLU networks

To verify that the accelerated learning we observed with curl descent in tanh networks is not restricted to sigmoidal activation functions, we replicated the experiments obtained for tanh networks in feed-forward architectures whose units employed rectified-linear (ReLU) activations. We used the same student-teacher set-up ($M = 100$ inputs, $N = 10$ hidden units), and identical parameters. The faster convergence effect on ReLU networks was smaller, hence the 40 random seeds for statistical significance. The results are shown in figure 10.

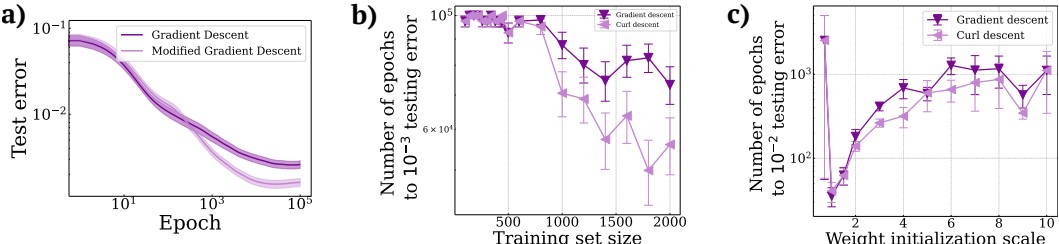

Figure 10: **Relu networks: curl descent leads to faster convergence in a broad parameter regime. a)** Test error for curl descent and gradient descent ($N_{\text{train}} = 1400$, weight initialization scale = 2; error bars indicate ± sem, averaged over 40 random seeds). **b)** Convergence speed of curl descent and gradient descent as a function of training set size (weight initialization scale = 2, $p < 0.05$ for $N_{\text{train}} \geq 1000$). **c)** Same as b) as a function of the weight initialization range ($N_{\text{train}} = 10000$, $p < 0.05$ for weight initialization scales $3, 4, 6, 7$ and $8$). Compute resources: 24 hours on 500 CPUs (local cluster).

# E   Faster convergence without the weight renormalization step

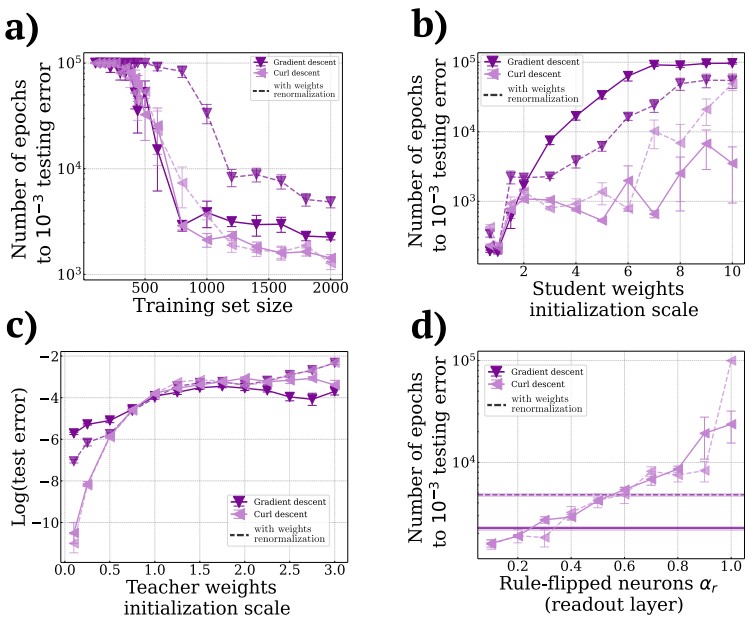

Figure 11: **Nonlinear networks: curl descent without renormalizing the weight matrices at each time step leads to faster convergence in a broad parameter regime. a)** Convergence speed of curl descent and gradient descent as a function of training set size (weight initialization scale = 2). **b)** Same as c as a function of the weight initialization range ($N_{\text{train}} = 10000$). **c)** Log of the test error after $20,000$ epochs as a function of the teacher weights initialization scale. **d)** Convergence speed as a function of the fraction of rule-flipped readout neurons. Compute resources: 12 hours on 500 CPUs (local cluster).

