# OpenReview forum: "Curl Descent : Non-Gradient Learning Dynamics with Sign-Diverse Plasticity"
_NeurIPS.cc/2025/Conference — NeurIPS 2025 spotlight_

### Official Review · Reviewer_mR3k · 2025-06-17

**Clarity:** 4
**Significance:** 3
**Originality:** 3
**Rating:** 5
**Confidence:** 3

**Summary:**

In this paper, the authors demonstrate that learning is still possible in neural networks in which the sign of synaptic plasticity for some neurons is flipped. They develop a formal theory that characterizes the curl dynamics induced by such sign flips, and characterize the effects of sign flips on the fixed point structure of learning dynamics. Lastly, the authors show that under some conditions the presence of sign flips can, counterintuitively, be beneficial for optimization.

**Questions:**

Major concerns:

1.	In section 2, it is unclear to me whether the curl descent analysis provided is useful, because unlike in the gradient descent framework in section 3, fixed points of the network are not well-characterized. This is relevant, because the Hebbian synaptic plasticity rule selected in Eq. 1 typically produces explosive weight dynamics if M_ij is positive, and vanishing dynamics if M_ij is negative. After all, if the plasticity rule is not stable or results in vanishing synaptic magnitudes, what is the value of this analysis?

2.	For Figure 5, the authors have provided a demonstration that flipping the sign of plasticity for one readout weight improves performance significantly in a student teacher formulation. However, this is really not enough to be able to make much of any claim about the effects of curl descent on learning dynamics under other conditions. For instance, what happens as the number of flipped signs increases? What happens if signs are flipped in the hidden layer instead? What happens if the network is trained on an even moderately more complex task (e.g. MNIST), or if more complicated network architectures are used (e.g. convolutional)? Because this claim is counterintuitive (and potentially very cool), more empirical evidence is required to support the claim; while I accept the intuition that curl descent could help networks avoid local minima or slow points, making such a method function adequately at scale may require additional tricks (e.g. taking an annealing-style/learning rate scheduled approach to curl descent).

3.	While the authors seem to assume that sign flips in plasticity exclusively induce non-gradient curl terms in weight dynamics, this is not the only possibility, and alternatives should be discussed. I see four alternative possibilities. First, and least interesting, the functional form of ‘optimal’ synaptic plasticity within a normative framework can change quite radically based on the biophysics of a neuron and the ‘experimental’ conditions under which plasticity is characterized, meaning that cell-type specific plasticity could simply be a due to evolutionary corrections for differences in the biophysics of different neuron subtypes. Second, the sign of synaptic plasticity has been shown to vary considerably based on the position of a synapse relative to the soma in pyramidal neurons (Froemke et al., 2010; Sjostrom & Hausser, 2006); several other theoretical frameworks have argued that this switch in the sign of plasticity is actually fully consistent with approximate gradient descent learning (Richards & Lillicrap, 2019). Third, neuromodulators have been show to induce flips in the sign of synaptic plasticity, and many studies have attributed this effect to a natural consequence of reinforcement learning (Frémaux & Gerstner, 2016). And lastly, cell-type specific variations in plasticity could be due to the optimization of local auxiliary objectives that effectively place constraints on the optimization landscape for other non-local optimization processes (e.g. inhibitory plasticity could simply be serving the purposes of local homeostatic normalization of network activity). Recent studies have shown that network stabilization can emerge as a consequence of a wide diversity of different plasticity updates (e.g. Confavreux et al., 2025).

Minor concerns:

1.	A recent paper (Shoji et al., 2024) has argued that a broad family of plasticity rules (even those with apparent curl) can be viewed as natural gradient descent if one chooses an appropriate timescale over which to average plasticity updates. While at face value this argument appears inconsistent with the conclusions of this paper, I actually think that they are fully consistent—either the plasticity covered in the current paper violates the constraints required for formal argument in the natural gradient study, or it may indeed be viewed as a form of natural gradient descent if averaged over an appropriate timescale. Regardless, the paper should be discussed because of their similar setup and differing conclusions; it would fit nicely into the ‘Natural gradients’ section.

2.	Why can the first term in Eq. 2 be viewed as the negative gradient of an energy function? It is not immediately clear to me why the expression written can be viewed as an energy function (or what assumptions are being made to make it an energy function).

3.	Is it possible to generalize the mathematical analysis here to perturbations more general than sign flips? It seems to me like this analysis ought to be able to be generalized to a much broader family of diagonal perturbation matrices without much change to the underlying math.

**Ethical Concerns:**

["NO or VERY MINOR ethics concerns only"]

**Final Justification:**

The authors addressed my concerns, by providing both additional empirical evidence to support their claims and additional explanations that should increase the clarity of the paper.

**Limitations:**

Yes

**Quality:**

3

**Strengths And Weaknesses:**

The paper is very clearly written, and quite a bit of effort is put into building the reader’s intuitions, which was much appreciated. It is also original, as far as I can tell. I think that this paper is very interesting, primarily because of the formal theory that the authors are able to develop to mathematically analyze learning dynamics in sign-flipped networks that learn via gradient-based signals—in my mind, this contribution alone is probably sufficient to warrant acceptance with some revisions, but other parts of the paper could be significantly improved. I think that the value of Section 2 is less clear, and that the authors could significantly improve the paper by clarifying a few points regarding that section (discussed below). I am also generally unconvinced by the results that curl descent can improve performance relative to gradient descent in more complicated networks or on more complicated tasks.

---

> ### Author Rebuttal · Authors · 2025-07-30
>
> We thank the reviewer for their constructive comments. We respond point-by-point below.
>
> # Major concerns
>
> ## 1. Curl descent analysis of Hebbian/anti-Hebbian plasticity
>
> The rationale for Section 2 is only to provide motivation for sign-diverse plasticity by showing how they can arise in biological neural networks, before moving to the more tractable teacher-student framework. We agree the rule in Eqn 1 would require additional terms to stabilize it, but this will not in general affect the curl terms.
>
> For example, one common way to stabilize weight dynamics is by adding a leak term of $-\gamma W$ to the plasticity rule. This is just the gradient of the Frobenius norm $-\frac{\gamma}{2} \Vert W \Vert_F^2$, and can be considered as a regularization added to the original objective defined by the rest of the plasticity rule (if such an objective exists).
>
> However, if we incorporate the term $-\gamma W$ to the plasticity rule, this will not affect whether or not the entire rule can be written as a gradient. As we showed in Appendix A.2, the Jacobian of the original rule in Eqn 1 is not symmetric when $M$ is sign diverse. A symmetric Jacobian is necessary for the plasticity rule to be a gradient system. Since the leak term itself has a symmetric Jacobian, adding it to Eqn 1 cannot symmetrize the entire plasticity rule’s Jacobian. The same logic holds for any stabilization term that is itself gradient-based.
>
> To avoid overburdening the reader, we had originally removed the leak term from Section 2 to focus only on the sign flips, but we acknowledge that this section may be confusing. In response to this reviewer’s feedback, as well as a suggestion from reviewer 2UnU, we will move the Hebbian/anti-Hebbian example to the appendix, incorporating the points above about the stabilization of the plasticity rule. We will keep only the EI network in the main text, and we will explicitly state that this section is only meant to motivate the sign flips before proceeding to the student-teacher framework that is used for the rest of the paper. Additionally, we will change the term ‘anti-Hebbian-like neurons’ for the more precise term 'rule-flipped neurons’.
>
> ## 2. Additional simulations for Figure 5
>
> Additional simulations were conducted for figure 5, to which we added two panels, one regarding the generalization of curl descent benefits to a broader range of architectural parameters, and the other regarding task complexity. Due to limited space, we redirect the reviewer to the answer to reviewer 35Tb under the section “Additional simulations for Figure 5”.
>
> We thank the reviewer for the excellent point about using tricks such as annealing to make curl descent more robust. We will add this to the Discussion.
>
> ## 3. Alternatives to non-gradient curl terms
>
> We fully agree that opposite-sign plasticity rules are not in themselves incompatible with gradient descent, and did not intend to suggest otherwise. Since gradient flow is the dominant normative framework for synaptic plasticity, we aimed to demonstrate how easily curl terms could arise through sign diverse plasticity rules. However, we acknowledge that a deeper discussion of these different possibilities would be useful.
>
> Below we respond to each of the four possibilities.
>
> * In the first case, in networks with defined cell types, gradient descent can lead to diversity in the plasticity rules as a result of that heterogeneity. The reviewer mentions differences in intrinsic biophysical properties of individual neurons, which would predict a strong correlation between cell type and plasticity rule. Another example is diversity in structural connectivity. For example, Pehlevan and Chklovskii 2015 have shown that Hebbian plasticity in structured EI networks can implement gradient descent on a similarity matching objective. In Appendix A.4 we demonstrate that these networks are able to implement gradient flow by choosing a specific architecture that nullifies potential curl terms.
>
> * The second case of distally-dependent plasticity also brings up the importance that learning rules correlated with structural parameters (here distance to the soma) do have in biological neural networks. While we do not implement such structurally dependent learning rules in this work, we believe that this would be a very interesting direction to explore and will include this in the discussion.
>
> * Regarding sign-flips induced by neuromodulators, we thank the reviewer for pointing us to this body of work. This could provide a mechanism for biological networks to adaptively tune their curl terms in order to escape saddles and shallow minima. This could be a fascinating direction to explore in future work and could provide an alternative to reinforcement learning based framework in Fremeux and Gerstner.
>
> * Finally, it is true that different plasticity rules could be considered as separate objectives (homeostatic vs. error minimization) that may be locally implemented in different cell types. However, in curl descent, the plasticity rules are not diverse in their functional form, but rather have opposite signs. The rule-flipped neurons are ascending the gradient, and are therefore adversarial rather than homeostatic (see also our reply to 2UnU above). However, sign flipped rules could implement a homeostatic minimization if combined with cell-type-specific differences in connectivity profile or biophysical property, as in the first case above.
>
> We will explicitly discuss these different possibilities where learning rules are flipped without causing curl terms in the Discussion section, along with the recommended citations.
>
> # Minor concerns
>
> ## 1. Shoji et al., 2024
>
> We thank the author for highlighting this work. We do cite this paper in Section 6 (Related work – Natural gradients), but we agree that it presents an interesting perspective that warrants more discussion. Shoji et al give a constructive proof that any learning rule which monotonically decreases a cost function after some time window implements a form of natural gradients, if we consider the integrated weight updates over this time window. In the case of curl descent, however, there may be regions of the parameter space where the learning dynamics are not monotonically decreasing, regardless of the time window considered. For example, in the toy model in Figure 1a, initializing within the heteroclinic orbits results in periodic weight dynamics, violating the core assumption of Shoji et al. In simulations in larger networks, we did not observe such non-decreasing loss curves (as long as only a minority of neurons in the readout layer were flipped), but we cannot rule out the existence of some weight initialization regimes that would lead to this. We will incorporate these points into the discussion on natural gradients.
>
> ## 2. Energy function in Eqn 2
>
> Here we simply meant that this term by itself is the gradient of an objective function. Specifically, this term can be written as $\mathbf{f}\mathbf{f}^\top \odot M = -\nabla_W \mathcal{L}(W)$, where $\mathcal{L}(W) = -\text{Tr}((\mathbf{f}\mathbf{f}^\top \odot M) W^\top)$. This loss is an inner product between $W$ and the matrix $(\mathbf{f}\mathbf{f}^\top \odot M) $. Therefore when $M = \mathbf{1} \mathbf{1}^\top$ (no flipped synapses), this term tends to align the recurrent weights to the input covariance matrix. We’ll clarify this point in the text.
>
> We also note that the term “energy function” may be misleading as it is often used to refer to gradient-based dynamics of the hidden unit activations (e.g., in Hopfield networks) rather than learning dynamics of $W$. To avoid misunderstanding, we will change this term simply to “objective function”.
>
> ## 3. Generalization to diagonal matrices
>
> Our theoretical analysis could indeed be generalized to a broader range of diagonal perturbations. Assuming such general diagonal perturbations will not change the derivations leading to Eqns 13 and 14 in the main text. However, the subsequent random matrix theory derivations will have to be adapted. This would require using the Cauchy and Blue transforms (Eqns 89 and 90 of the supplementary materials) of a general diagonal matrix for perturbations of the hidden layer. Similarly, in the case of perturbations of the readout layer one could derive the support boundaries using the method of Rao & Edelman, 2006 (“The polynomial method of random matrices”), but here we expect a tedious calculation.

---

> > ### Comment · Reviewer_mR3k · 2025-08-05
> >
> > Thank you for an excellent response--your comments have addressed my primary concerns, so I increased my score accordingly.

---

### Official Review · Reviewer_35Tb · 2025-06-27

**Clarity:** 3
**Significance:** 4
**Originality:** 3
**Rating:** 5
**Confidence:** 3

**Summary:**

This paper introduces curl dynamics as an extension to standard gradient descent optimization, drawing inspiration from biological neural networks. Using a toy model, the authors offer clear intuition behind the proposed approach. They then analyze the stability of curl descent within a teacher-student framework using linear transformation networks, supporting their theoretical findings with numerical experiments.

**Questions:**

I noticed that the test error heat map (Fig. 4a) appears to differ significantly from the theoretical prediction (Fig. 2, bottom). In contrast, the max error heat map seems to align more closely with the theoretical results based on random matrix theory. I may be misunderstanding something, but the discrepancy between the numerical results and theoretical predictions does not seem to be fully addressed in the paper. Could the authors clarify this point?

Additionally, what is the origin of the U-shaped pattern observed in the test error heat map (Fig. 4a)? This U-shape appears even more pronounced when using the nonlinear tanh activation function, as shown in Supplementary Material (page 14, Fig. 3a). Is there an underlying theoretical explanation for this behavior?

For reference, the importance of curl dynamics in neural networks has been previously highlighted—for example, in Yan et al., Nonequilibrium landscape theory of neural networks, PNAS 110(45): E4185–E4194 (https://doi.org/10.1073/pnas.1310692110).

**Ethical Concerns:**

["NO or VERY MINOR ethics concerns only"]

**Final Justification:**

After reviewing the responses and related discussions, I have decided to maintain my original rating of 5 (accept).

**Limitations:**

Yes

**Quality:**

3

**Strengths And Weaknesses:**

Strength:
The paper offers a solid theoretical foundation for curl descent, employing analytical techniques based on a toy model and random matrix theory to provide clear rationale and insight.

Weakness:
While the work serves as a strong proof of concept, it falls short by not demonstrating the practical effectiveness of curl descent in real-world deep learning scenarios with more realistic datasets.

---

> ### Author Rebuttal · Authors · 2025-07-30
>
> We thank the reviewer for their constructive comments. We respond point-by-point below.
>
> ## 1. Mismatch between test error heat map and theoretical prediction:
>
> Indeed, Figure 4a shows that the test error is low even when the analytical theory predicts that the solution manifold should become unstable. We were puzzled as well by this result, which led us to try initializing the weights very close to (but not directly on) the solution manifold, to test whether it was in fact numerically stable.
>
> We quantified numerical stability using the maximum error along the trajectory when initialized near the solution manifold. Recall that the error is zero directly on the solution manifold. Therefore, a max error near zero means that the trajectory converged to the solution manifold (thus it is numerically stable). A large max error indicates that the trajectory climbed up the gradient, leading to a transient increase in the error.
>
> These results are shown in Figure 4b, and precisely match the prediction from the analytical stability analysis. Together, Figure 4a and 4b demonstrate that the solution manifold does indeed lose stability as predicted by the theoretical results. However, this transition leads not to chaotic weight dynamics (as in Figure 3a), but rather towards other low-error solutions.
>
> We will rework the text describing Figure 4 to better explain why this mismatch occurs, and why it is in fact consistent with the analytical theory.
>
> ## 2. U-shaped pattern in the test error heat map:
>
> To produce Figure 4a, we trained networks with a fixed 220 total number of neurons, varying the compression ratio $c$. We observed that the number of epochs to convergence increased with $c$ for both gradient descent and curl descent. In the case of curl descent, we noticed that as the curl terms were strengthened by increasing the fraction of rule flipped neurons, the transient learning dynamics lasted longer before settling into a low-error region. The red U-shape region of Figure 4a therefore corresponds to simulations that reached the $10^5$ epochs before the transient dynamics ended. Increasing the number of epochs effectively reduces the size of this U-shaped pattern.
>
> Interestingly, the bottom-right side of this U-shaped pattern (where $c=10$ and the fraction of rule-flipped  neurons = 0.1) delimits a region where the transient dynamics appear to be shorter. In this region, curl descent actually performs better than gradient descent in both the linear and the nonlinear cases (with the nonlinear case being the focus of figure 5). Unfortunately, we do not yet have a theoretical explanation for this phenomenon, but we will raise this interesting observation in the Discussion.
>
> ## 3. Nonequilibrium landscape theory of neural networks
>
> We thank the reviewer for this citation, which we originally missed. We will incorporate it into Section 6 – Related work.
>
>
> ## Additional simulations for Figure 5
>
> We fully agree that the empirical evidence presented in this work does not demonstrate the generalization of curl descent gains to more complex architectures or tasks. The primary contribution of this paper is to give a first treatment of non-gradient dynamics, in an analytically tractable setting. Our results therefore show that within this setting, curl descent can enhance learning within a limited region of structural parameter space that can be analytically identified. To date, we lack analytical results that would guide the correct parameterization of more complex networks on real-world datasets.
>
> However, as suggested by reviewers mR3k and 2UnU, we will improve Figure 5 through additional empirical results regarding the effect of curl descent dynamics under other conditions, while still operating within the student-teacher setting described in our work. We conducted further numerical analyses on two recurring concerns from the reviewers’ feedback: the generalization of curl descent benefits to other architectural parameters and task complexity.
>
> ### A. Architectural parameters
>
> We conducted new simulations to investigate the effect of increasing the fraction of rule-flipped rule neurons in the readout layer for three compression ratios (0.1, 1 and 10). We found that the benefits of curl descent, as well as the range of rule-flipped neurons fraction improving learning, were more pronounced in networks with high compression.
>
> Regarding the introduction of rule-flipped rule neurons in the hidden layer, we did not find any structural parameters that lead to improved performance compared to gradient descent, consistent with our analytical results in linear networks (Figure 3a)
>
> ### B. Modulating task complexity
>
> We have added new simulations where we parametrically increase the complexity of the task by expanding the teacher’s initialization range. This is known to increase the complexity of the teacher representation and, consequently, the difficulty of the task the student must learn (Poole et al., 2016). These simulations reveal a threshold initialization range above which there is a sharp drop in convergence speed for both curl descent and gradient descent, as the dynamics in either case are stuck in a saddle. Above this threshold, both rules converge at the same (slow) speed. Below this threshold, curl descent converges faster than gradient descent.

---

> > ### Comment · Reviewer_35Tb · 2025-08-04
> >
> > Thank you for your response. After reading the other reviews and your rebuttal, I've no further questions.

---

### Official Review · Reviewer_2UnU · 2025-07-02

**Clarity:** 3
**Significance:** 4
**Originality:** 3
**Rating:** 5
**Confidence:** 4

**Summary:**

This paper is a normative study on synaptic plasticity in neural networks. Normative approaches have mostly focused on learning rules that minimize a given cost function (i.e. that follow a gradient). Conversely, the plasticity rules that are not derived from a cost function are typically considered from a purely empirical or numerical point of view. Here, the authors provide an analytical perspective on the latter group of rules. They first prove that “sign diversity” (in the plasticity rule: co-active Hebbian + anti-Hebbian plasticity; or in the neuron types: excitatory and inhibitory neurons) generate learning dynamics that do not follow any gradients.
The trick to nevertheless provide a normative treatment of such systems is to use a teacher-student framework in linear feedforward networks with a proportion of synapses doing gradient ascent while others do gradient descent. The learning dynamics can then be broken down into a classic gradient-descent component and additional “curl” terms that affect the learning dynamics depending on parameters such as the network architecture and the proportion of flipped synapses. The authors show that curl terms can change which solution from the gradient-descent dynamics is reached, sometimes speeding up learning.

**Questions:**

1/ Each plasticity rule/cell type could be understood to optimize its own gradient/objective. Even though the whole system doesn’t optimize a single objective, could it still be understood as some competition across neurons for different cost functions? This would somewhat nuance your claim challenging the prevailing view of gradients for synaptic plasticity.

2/ Over-focus on one specific instantiation of Hebbian and anti-Hebbian learning In section 2. What about other rules? For instance variations of hebbian/anti hebbian. I don’t think you need to add anything in main, but maybe point to a supplementary paragraph that suggests that the findings are not overly rule-specific. Right now, this makes your results appear to have a narrower scope than they surely have.

3/ Your motivation for adding different plasticity rules in the same network are cell types, which are typically defined by their connectivity patterns to other cell types and their unique cellular properties. Your model is thus making a rather strong assumption of only changing the rule. I understand an analytical treatment of such networks with more complex initial connectivity is out of scope, but maybe you could provide some numerics? Another way of asking the same question is: What if the identity of the flipped synapses was not random: can you show further/more robust benefits of curl terms in such networks? You could take inspiration from/recover classical synapse types connectivities? (local PV/SOM interneurons for instance)

4/ To make this work more impactful and strengthen the claim that curl terms are more than an “approximation error” to gradient descent and deserve proposer attention in the future, I think that figure 5 needs to be expanded. I understand this is out of scope/hard to do analytically, but some additional simulations would be welcome. E.g. recurrent networks/a more “real-world” task than the teacher student framework.

**Ethical Concerns:**

["NO or VERY MINOR ethics concerns only"]

**Final Justification:**

Overall, the authors have addressed my remaining concerns (clarity, discussion and additional experiments). I believe this is a high-quality contribution to the field and recommend acceptance.

**Limitations:**

Yes

**Quality:**

4

**Strengths And Weaknesses:**

Overall, I found this paper clear, well-explained and nicely presented. The model chosen is simple, at the limit of being simplistic, but just complex enough to obtain interesting conclusions about the problem at hand. As such I find it a very elegant and important work in the current scientific scene around synaptic plasticity, although it could be attempting to formulate more precise predictions towards the end.

---

> ### Author Rebuttal · Authors · 2025-07-30
>
> We thank the reviewer for their constructive comments. We respond point-by-point below.
>
> ## 1. Competition across neuron-specific cost functions
>
> This is a great point and the intuition is absolutely correct: each rule-flipped neuron is individually trying to climb the gradient, so in a sense they do have their own cost function, which is minus the mean squared error. These neurons can therefore be interpreted as not just competitive, but adversarial. With that perspective, we find it all the more intriguing that not only are the learning dynamics robust to such adversarial neurons, but that these adversarial neurons can even speed learning in certain situations.
>
> That adversariality makes curl descent different in spirit from a network in which two pools of neurons have competing cost functions that are separately normative (e.g., error minimization and homeostatic regulation). In other words, it isn’t a case of finding a compromise between slightly misaligned objectives: in curl descent, the neuron-specific objectives are fundamentally incompatible.
>
> Thus, while curl descent requires  computing gradients, it is not implementing gradient descent, not even for the individual populations. By having two populations of neurons with opposite cost functions,  the two cost functions can’t both decrease.
>
> We will add this important and nuanced point to the discussion, including the adversarial interpretation of curl descent dynamics, and the clarification that curl descent challenges *gradient descent* as a unified framework for learning, but still relies on *gradient computation*.
>
>
> ## 2. Other learning rules
>
> We agree that this section was overly focused on a specific form of Hebbian/anti-Hebbian plasticity. To give the simplest example of where non-gradient terms may arise, we used the product of pre- and post-synaptic activities which is common across variants of Hebbian plasticity. As suggested by the reviewer, it is straightforward to extend our analysis to these variants. One example, pointed out by Reviewer mR3k below, is adding additional terms to avoid weights decaying to zero or exploding to extreme values (e.g., in Oja’s rule, BCM, covariance or homeostatic rules). In our response to their comment, we provide an example in which including a leak term to stabilize the weights can be seen as a regularization term added to an objective function, and therefore does not affect the presence of curl terms. The same reasoning applies to other variants of Hebbian plasticity, which are often explicitly derived from the gradient of an objective.
>
> We acknowledge that section 2 in its current form, and the subsequent use of the term ‘Anti-Hebbian-like’ neurons in following sections, can be overly simplifying. We will address this by moving the ‘Hebbian/anti-Hebbian networks’ paragraph to supplementary materials (only focusing on the EI network example in Section 2), with an additional paragraph to explain how it can be extended to other variants of Hebbian plasticity. We will also change terminology from “anti-Hebbian-like” neurons to “rule-flipped  neurons”.
>
> ## 3. Cell types with non-random patterns
>
> The reviewer is right that assigning random rule flipped rule neurons is an important assumption in the paper. If the cell types are defined so that their plasticity rules *and* their structural connectivity is different, this can significantly affect the results. In particular, we showed an example in Appendix A.4 that correlating the plasticity rule to connectivity motif can result in gradient flow, if the structure is carefully chosen so that the curl terms are effectively nullified. This is the case in a previous proposed normative model of EI networks learning PCA (Pehlevan et al., 2015), which was the basis for that analysis.
>
> The reviewer instead raises an interesting point that we may be able to use this same idea  to enhance the curl terms, rather than nullifying them. This is indeed possible, although it would be challenging in the analytical framework of the paper. As we saw from our stability analysis, the architecture plays a crucial role in determining whether the curl terms will improve or impair learning performance, and a numerical treatment is unlikely to work without any theoretical guidance. A theoretical application to cortically defined structural connectivities will definitely be interesting but will require a new analytical framework (e.g., recurrent). We will add this point in the Discussion as both being a caveat of the work, and a potential future direction to find more robust curl terms.
>
> ## 4. Expansion of Figure 5
>
> We fully agree that the empirical evidence presented in this work does not demonstrate the generalization of curl descent gains to more complex architectures or tasks. The primary contribution of this paper is to give a first treatment of non-gradient dynamics, in an analytically tractable setting.  Our results therefore show that within this setting, curl descent can enhance learning within a limited region of structural parameter space that can be analytically identified. To date, we lack analytical results that would guide the correct parameterization of more complex networks on real-world datasets.
>
> However, as suggested by reviewers mR3k and 2UnU, we will improve Figure 5 through additional empirical results regarding the effect of curl descent dynamics under other conditions, while still operating within the student-teacher setting described in our work. We conducted further numerical analyses on two recurring concerns from the reviewers’ feedback: the generalization of curl descent benefits to other architectural parameters and task complexity.
>
> ### A. Architectural parameters
>
> We conducted new simulations to investigate the effect of increasing the fraction of rule-flipped rule neurons in the readout layer for three compression ratios (0.1, 1 and 10). We found that the benefits of curl descent, as well as the range of rule-flipped neurons fraction improving learning, were more pronounced in networks with high compression.
>
> Regarding the introduction of rule-flipped rule neurons in the hidden layer, we did not find any structural parameters that lead to improved performance compared to gradient descent, consistent with our analytical results in linear networks (Figure 3a)
>
> ### B. Modulating task complexity
>
> We have added new simulations where we parametrically increase the complexity of the task by expanding the teacher’s initialization range. This is known to increase the complexity of the teacher representation and, consequently, the difficulty of the task the student must learn (Poole et al., 2016). These simulations reveal a threshold initialization range above which there is a sharp drop in convergence speed for both curl descent and gradient descent, as the dynamics in either case are stuck in a saddle. Above this threshold, both rules converge at the same (slow) speed. Below this threshold, curl descent converges faster than gradient descent.

---

> > ### Comment · Reviewer_2UnU · 2025-08-04
> >
> > I thank the authors for their detailed response. This has increased my confidence in my (already favourable) score.

---

### Official Review · Reviewer_DQUi · 2025-07-03

**Clarity:** 2
**Significance:** 2
**Originality:** 3
**Rating:** 4
**Confidence:** 2

**Summary:**

The paper introduces Curl Descent, a theoretical framework for analyzing and leveraging non-gradient learning dynamics induced by biologically plausible, sign diverse plasticity rules. By decomposing learning updates via the Helmholtz decomposition into gradient-conforming and curl components, the authors formalize how heterogeneous synaptic plasticity; specifically the coexistence of Hebbian and anti-Hebbian rules: generates structured, controllable non-gradient dynamics. Using random matrix theory and explicit Jacobian spectral analysis, the paper quantitatively characterizes how the magnitude of these curl terms governs both the stability of learning trajectories and the geometry of the solution manifold. The results reveal that expansive architectures, where hidden layers have greater dimensionality than inputs, can inherently buffer against curl-induced instability, while networks operating near critical curl magnitudes exhibit rich, nonlinear learning dynamics.
The paper demonstrates that curl components are not universally detrimental to learning. Through an analytically tractable two-neuron system and larger simulations, the authors show that moderate curl magnitudes can facilitate beneficial dynamics, such as detours through unstable regions of parameter space that ultimately accelerate convergence or improve escape from saddle points. This phenomenon is particularly evident in non-linear networks, where introducing controlled curl terms reduces the number of training epochs required to reach low error levels, outperforming strict gradient descent under certain conditions.

**Questions:**

a. How does the interplay between the network’s expansion ratio (c = M/N) and the magnitude of the curl component quantitatively affect the phase boundaries for stable versus unstable learning dynamics in the random matrix regime, and can this relationship be generalized beyond the specific student-teacher model analyzed?

**Ethical Concerns:**

["NO or VERY MINOR ethics concerns only"]

**Final Justification:**

The authors have clarified my questions and concerns, and hence it is a favorable score.

**Limitations:**

yes

**Paper Formatting Concerns:**

Nope

**Quality:**

3

**Strengths And Weaknesses:**

Strength:
This work presents a well constructed and theoreticaly rigorous framework for understanding how biologically motivated, sign diverse plasticity rules induce curl components within learning dynamics. Through a principled application of Helmholtz decomposition, the authors distinguish between gradient conforming and non-gradient components of weight updates, providing a clear mathematical basis for analyzing non-gradient learning dynamics. The use of random matrix theory to characterize the stability properties of these dynamics, particularly through explicit derivations of Jacobian spectra and phase boundaries, reflects a commendable level of analytical depth. These results give concrete, testable predictions about how curl magnitude, network architecture, and plasticity sign diversity interact to govern learning behavior.
By linking curl dynamics to well-established biological principle; such as Dale’s law and empirical findings on heterogeneous long-term potentiation and depression—the authors successfully ground their mathematical results in experimentally observable phenomena. The inclusion of toy models, such as the analytically tractable two-neuron system, effectively conveys key intuitions regarding how curl components can introduce detours in parameter space that initially destabilize but ultimately accelerate convergence. These insights challenge prevailing assumptions in machine learning that gradient-following updates are strictly necessary for efficient learning and open promising avenues for exploring biologically inspired, non-gradient learning rules in artificial systems.

Weakness:
While the theoretical framework is compelling, certain technical claims, notably those regarding the uniqueness of the gradient term within the Helmholtz decomposition and the determinant-based stability conditions for larger networks, are only partially elaborated or deferred to supplementary material. Although the supplemental derivations appear sound upon spot-checking, presenting these results more completely within the main text, or providing an accessible computational (jupyter) notebook, would greatly improve transparency and facilitate verification by others.
While the simulation results qualitatively support the theoretical predictions, the empirical scope remains narrow, focusing primarily on small-scale synthetic tasks without extending to real-world benchmarks or diverse architectural settings. The absence of classification tasks, recurrent models, or larger datasets makes it difficult to assess how robust or generalizable the observed benefits of curl dynamics truly are.Lack of an accompanying, open source code release and incomplete reporting of experimental details;such as the implementatino details. Addressing these aspects would significantly improve the paper’s reproducibility and ensure that its insights are more easily transferable to practical machine learning and neuroscience research contexts.

---

> ### Author Rebuttal · Authors · 2025-07-30
>
> We thank the reviewer for their constructive comments. We respond point-by-point below.
>
> - ## Uniqueness of the gradient term:
>
> We emphasize that the Helmholtz decomposition is not used explicitly in our work. We mention the decomposition as an analogy for the existence of both gradient and non-gradient terms in the dynamics. We prove that curl descent dynamics is a non-gradient system without explicitly deriving a decomposition into a gradient field and rotational field. As such, we make no claims on the uniqueness of any gradient or non-gradient terms, and only require that the non-gradient term is non-zero.
>
> - ## Stability conditions for large $N$:
>
> For the stability conditions for large networks, we first derive the characteristic polynomial of the Jacobian of the curl descent learning dynamics, which we evaluate along the solution manifold. The eigenvalues of the Jacobian matrix of a dynamical system evaluated at a certain point are informative about the local stability at this point: if all eigenvalues’ real parts are negative, then the point is stable, otherwise it is unstable. We therefore use random matrix theory to determine when the spectral support includes positive values (in the limit that $N \to \infty$ and assuming i.i.d. Gaussian $W_1$, $W_2$). We do this by considering two cases: sign diversity in the hidden layer (i.e. in $D_1$) or in the readout layer (i.e. in $D_2$). In the readout layer, we use tools from free probability theory to derive the support boundaries of the spectral distribution as the roots of a polynomial that can be evaluated numerically. In the hidden layer, we can directly apply the method of Kumar and Sai Charan, 2020 which directly provides the expression of the polynomial whose roots determine the spectral support’s boundaries. All derivations are in the appendix due to limited space in the main text. However, we sympathize with the reviewer’s concern that the basis for these derivations may be obscure. We will therefore include a brief sketch of the derivation in Section 4.2, and we will include additional details in the Appendices to walk the reader carefully through the derivations. Additionally, we will provide a jupyter notebook comparing the empirically obtained spectra of the random matrices to the theoretical predictions.
>
> - ## Generalization of curl descent benefits:
>
> We fully agree that the empirical evidence presented in this work does not demonstrate the generalization of curl descent gains to more complex architectures or tasks. The primary contribution of this paper is to give a first treatment of non-gradient dynamics, in an analytically tractable setting.  Our results therefore show that within this setting, curl descent can enhance learning within a limited region of structural parameter space that can be analytically identified. To date, we lack analytical results that would guide the correct parameterization of more complex networks on real-world datasets.
>
> However, as suggested by reviewers mR3k and 2UnU, we will improve Figure 5 through additional empirical results regarding the effect of curl descent dynamics under other conditions, while still operating within the student-teacher setting described in our work. We conducted further numerical analyses on two recurring concerns from the reviewers’ feedback: the generalization of curl descent benefits to other architectural parameters and task complexity.
>
> ### 1. Architectural parameters
>
> We conducted new simulations to investigate the effect of increasing the fraction of rule-flipped rule neurons in the readout layer for three compression ratios (0.1, 1 and 10). We found that the benefits of curl descent, as well as the range of rule-flipped neurons fraction improving learning, were more pronounced in networks with high compression.
>
> Regarding the introduction of rule-flipped rule neurons in the hidden layer, we did not find any structural parameters that lead to improved performance compared to gradient descent, consistent with our analytical results in linear networks (Figure 3a)
>
> ### 2. Modulating task complexity
>
> We have added new simulations where we parametrically increase the complexity of the task by expanding the teacher’s initialization range. This is known to increase the complexity of the teacher representation and, consequently, the difficulty of the task the student must learn (Poole et al., 2016). These simulations reveal a threshold initialization range above which there is a sharp drop in convergence speed for both curl descent and gradient descent, as the dynamics in either case are stuck in a saddle. Above this threshold, both rules converge at the same (slow) speed. Below this threshold, curl descent converges faster than gradient descent.
>
> - ## Transparency and open source code:
>
> We wholeheartedly agree with the reviewer’s concerns regarding transparency and reproducibility. We will make all code used for the simulations publicly available on GitHub, including in the form of jupyter notebooks.
>
> - ## Generalizations of the phase boundaries:
>
> We used the approach described above to solve for the phase boundary between stable vs. unstable learning dynamics, as a function of $c$ and the number of sign flips (which is a proxy for the curl component).  We focused on two-layer feedforward student-teacher networks as they are a widely studied framework for mathematically tractable analyses of gradient flow. This allowed us to derive precise theoretical insights as a first step towards understanding non-gradient learning dynamics. Our analyses could in principle be extended to other architectures, including deep networks or with convolutional layers, although the derivation of the spectral support will surely be more onerous. Extensions to recurrent architectures are more challenging, as the interplay between recurrent dynamics and learning dynamics could generate new phenomena that are beyond the scope of this study. Nevertheless we acknowledge that our study is just a first step to understanding non-gradient terms. We will highlight these limitations in the discussion as promising directions for future work.

---

> > ### Comment · Reviewer_DQUi · 2025-08-05
> >
> > Thank you for the detailed clarifications, the paper is much more clear now. I have updated my score accordingly.

---

### Note · Authors · 2025-08-16

We thank the reviewers once again for their thoughtful and constructive feedback. We are encouraged by their appreciation of the "rigor", "elegance", and "originality" of our theoretical framework. We particularly appreciate the questions raised regarding natural gradients, non-equilibrium dynamics, synaptic plasticity, and learning rules, as they opened up intriguing connections to the broader literature. We thus hope that our work will provide a strong theoretical foundation for curl descent learning dynamics.

---

### Decision · Program_Chairs · 2025-09-17

**Decision:**

Accept (spotlight)

**Comment:**

The paper is motivated by biologically plausible training of neural networks using non-gradient based rules.  It introduces curl dynamics associated with sign flips and it develops an interesting theory, based on a teacher-student framework, that analyzes the stability of
the learning dynamics. The authors provide experimental evidence showing that, in certain nonlinear architectures, their so called curl descent can even accelerate learning compared to gradient descent.

As all reviewers agree, the paper introduces an original idea with a very clear motivation and exposition. Many attendees in the conference especially from the field of computational neuroscience will find this work very interesting.